

# Low-Level, Liquid-Only and Mixed-Phase Cloud Identification by Polarimetric Lidar

Robert A. Stillwell[1], Ryan R. Neely III[2], Jeffrey P. Thayer[1], Matthew D. Shupe[3,4], and Michael O'Neill[3]

[1]Aerospace Engineering Sciences, University of Colorado at Boulder, ECNT 320, 431 UCB, University of Colorado, Boulder, CO 80309.

[2]National Centre for Atmospheric Science and the School of Earth and Environment, University of Leeds, Leeds, LS2 9JT, United Kingdom.

[3]Cooperative Institute for Research in Environmental Sciences, University of Colorado at Boulder, 216 UCB, University of Colorado, Boulder, CO 80309.

[4]Earth System Research Laboratory, National Oceanic and Atmospheric Administration, 325 Broadway, Boulder, CO 80309.

*Correspondence to:* Robert A. Stillwell (robert.stillwell@colorado.edu)

**Abstract.** The measurement of low-level, liquid-only and mixed-phase clouds in the polar regions is a necessary building block to understand the regional surface energy and mass budgets over ice sheets. The unambiguous retrieval of cloud phase from polarimetric lidar observations is dependent on the assumption that only cloud scattering processes alter the transmitted polarization. However, due to clouds varying in range, optical depth, and scatterer size and shape, most atmospheric lidar systems must observe high dynamic ranges in scattered signal strengths. Depending on the polarization component measured, these signals can far exceed the linear range of a detection system. Thus, due to the high optical thickness and predominately low-lying nature of liquid-only and mixed-phase clouds in the polar regions, relative to ice only clouds, a systematic overestimate of the traditional depolarization ratio, which uses the co-polarized and cross-polarized signals, can occur due to the large dynamic range signals. For both liquid-only and mixed-phase clouds, this results in a misidentification of liquid water in clouds as ice, which has broad implications on evaluating surface energy budgets. The Clouds Aerosol Polarization and Backscatter Lidar (CAPABL) at Summit, Greenland employs multiple planes of linear polarization, and photon counting and analog detection schemes, to self evaluate, correct, and optimize signal combinations to improve cloud classification. For example, an examination of observations of liquid-only and mixed-phase clouds at Summit shows as much as a 2 kilometer offset in median cloud height of those identified as liquid due to a systematic bias in photon counting signals. At a constant altitude, more than 94% of the liquid pixels identified with analog signals can be misidentified as ice with photon counting signals. This results in a possible error of fractional occurrence of cloud liquid of approximately 30%. It is shown that by observing polarization planes that are non-orthogonal, the dynamic range of observed signals is reduced, the coverage of the expected signal dynamic range is increased, and more linear response can be captured. Using non-orthogonal polarization observations is shown to enhance measurement sensitivity increasing the effective sampling range for CAPABL by as much as 18% or approximately 1.5 km.





# 1  Introduction

The polar regions have recently come into focus as a critical building block to understand the climate system. Changing Arctic conditions lead to many changes in regional surface energy and mass budgets, which have a profound impact on humans outside the region (Curry et al., 1996; Hansen et al., 2011). Locked within the Greenland Ice Sheet (GrIS) is the the potential

for sea level rise on the order of $7\,m$ (Gregory et al., 2004), of which approximately $25\,mm$ has already been contributed from 1900 to present with an increased rate of mass loss in recent years (Kjeldsen et al., 2015). Several studies have linked variability of the surface energy and mass budgets to cloud properties and in particular low-level, liquid-only and mixed-phase[1] clouds (Bennartz et al., 2013; Sherwood et al., 2014; Miller et al., 2015; Tan et al., 2016). Shupe and Intrieri (2004) note that modeling is the most practical method to understand widespread climate change, which is especially true in the polar regions

where dense observations are inhibited by the remote and extreme conditions found there. While the climate is sensitive to cloud macro and microphysical properties, substantial gaps are present in understanding of fundamental cloud processes due to a limited set of cloud observations to which model results may be compared (Curry et al., 1996; Cesana et al., 2012; Morrison et al., 2012; Bennartz et al., 2013; Van Tricht et al., 2016).

Understanding the nature of liquid-only and mixed-phase clouds is important for understanding the polar surface energy

budget. Mixed-phase clouds show remarkable persistence in the Arctic even though the liquid phase is colloidally unstable, possibly persisting for days to weeks given the correct synoptic conditions (Shupe et al., 2006). Furthermore, though liquid-only and mixed-phase clouds can be found up to heights of approximately 6 km AMSL in the polar regions, they have been found by many to be predominately low-lying with high optical thickness (note that in this manuscript, high is taken relative to ice-only clouds existing in the same region and not to liquid clouds existing in the mid-latitude or tropical regions) (Curry

et al., 1996; Intrieri et al., 2002; Turner, 2005; Shupe et al., 2006; de Boer et al., 2009; Shupe, 2011; Shupe et al., 2013). Such characteristics make these clouds particularly hard to measure accurately from both the ground and space. Shupe et al. (2006) further notes that mixed-phase clouds are an understudied component of global cloudiness resulting in their poor representation in models at all scales, a finding supported by others including Cesana et al. (2012); Pithan et al. (2014); Kay et al. (2016). The focus of this paper is the interpretation of ground based polarimetric lidar measurements of Arctic liquid-only and mixed-phase

clouds and addressing systematic biases that improve their proper identification.

Polarimetric lidar data is particularly useful for cloud and aerosol studies to determine properties such as cloud phase, cloud base height, particle orientation, extinction, and for broad aerosol classifications (Schotland et al., 1971; Measures, 1984; Sassen, 1991; Kaul et al., 2004; Fujii and Fukuchi, 2005; Weitkamp, 2005; Freudenthaler et al., 2009; Hayman and Thayer, 2012; Groß et al., 2015). Its utility can be enhanced by using complementary measurements that grant a more complete

perspective such as cloud radars, microwave radiometers, and radiosondes as done for programs like the Surface Heat Budget of the Arctic Ocean (SHEBA) (Shupe et al., 2006), the Department of Energy atmospheric observatories (Verlinde et al., 2016), and Mixed Phase Arctic Clouds Experiment (MPACE) (Verlinde et al., 2013). Despite its utility, polarimetric lidar has

---

[1]This work uses the definition of mixed-phase presented by Shupe et al. (2008) where a mixed phase cloud is defined as a cloud system containing both liquid and ice water that interact via microphysical processes. The complete system must contain both liquid and ice water but no requirement is made on the exact location or quantity of either phase.

limitations, among them is the stringent requirement of linear signal operation over a large dynamic range. If not properly designed or considered, measurements can be misinterpreted casting doubt on critical measurements like cloud phase. These measurement errors in turn introduce unquantified error into model results which are used to study key processes. Observations by lidar of polar liquid-only and mixed-phase clouds in particular are challenging due to their high optical thicknesses relative

to ice-only clouds and low-lying altitude which demands large system dynamic ranges.

This work focuses on lidar measurements made at Summit, Greenland ($72°35'46.4"N$, $38°25'19.1"W$, $3212\ m\ asl$) as part of the Integrated Characterization of Energy Clouds Atmospheric State and Precipitation at Summit (ICECAPS) program outlined by Shupe et al. (2013). The measurements to be presented are taken from the Clouds Aerosol Polarization and Backscatter Lidar (CAPABL), which was originally designed to measure polarization properties of clouds with emphasis on identifying

preferentially oriented ice crystals and cloud phase (Neely et al., 2013). Analysis of five years of data observed by CAPABL has highlighted several uncertainties and biases that cause errors in the interpretation of geophysical retrievals of cloud phase, primarily caused by the system limitations to adequately handle the observable dynamic range in backscattered signals from clouds. The goal of this paper is to definitively describe these errors, using CAPABL as an example, with emphasis on improving the geophysical interpretation of lidar measurements.

The outline of this paper is as follows. The polarization theory used, upon which the retrievals within CAPABL's automatic processing are based, is stated in Sect. 2. An overview of the CAPABL system is provided in Sect. 3 with an emphasis on the updates that have been made to the instrument since its introduction by Neely et al. (2013). An overview of the data processing is provided in Sect. 4 with emphasis on geophysical retrievals and potential errors caused by limited signal dynamic range. Relevant cloud statistics for a 4 month period are compiled in Sect. 5. A comparison of results focusing on the errors of

geophysical cloud property estimates resulting from data misinterpretation is given in Sect. 6. Finally this paper concludes in Sect. 7 with a summary and recommendations for general cloud observations via lidar and suggestions for additional work to further improve polarimetric lidar retrievals of polar liquid-only and mixed-phase clouds.

## 2 Polarization theory

### 2.1 CAPABL's Polarization Retrievals and Mueller Formalism

Polarimetric lidar leverages the vector nature of light to more completely describe scattering. Using a vector description of light allows one to describe scatterers by how they alter polarization states of light as well as how much energy is redirected. Hayman and Thayer (2012) use polar decomposition of Mueller matrices to define the Stokes vector lidar equation (SVLE), which links transmitted and received polarization states of light to physical attributes of the scatterers. This equation forms the basis of CAPABL's polarization retrievals and is given in Eq. 1

$$\bar{N}(R) = \bar{\bar{O}}\bar{\bar{M}}_{R_x}(\bar{k}_s)\left[\left(G(R)\frac{A}{R^2}\Delta R\right)\bar{\bar{T}}_{atm}(\bar{k}_s, R)\bar{\bar{F}}(\bar{k}_i, \bar{k}_s, R)\bar{\bar{T}}_{atm}(\bar{k}_i, R)\bar{\bar{M}}_{T_x}(\bar{k}_i)\bar{S}_{T_x} + \bar{S}_B(\lambda_{R_x})\right] \qquad (1)$$



where $\bar{N}$ is vector of photon counts for each polarization channel as a function of range, $R$, $\bar{\bar{O}}$ is the observation matrix describing each polarization observation channel, $\bar{\bar{M}}_{T_x}$ and $\bar{\bar{M}}_{R_x}$ are the Mueller matrices describing the transmitter and receiver which are functions of the incident and scattered wave vector $\bar{k}_i$ and $\bar{k}_s$, respectively, $G$ is the physical overlap function of the transmitter and receiver, $A$ is the telescope area, $\Delta R$ is the range resolution of the counting system, $\bar{\bar{T}}_{atm}$ is the

one way transmission Mueller matrix either between the transmitter and the scatterer or between the scatterer and the receiver, $\bar{\bar{F}}$ is the scattering phase matrix which is a function of both transmitted and received wave vectors and range, $\bar{S}_{T_x}$ is the Stokes vector of the light from the laser source, and $\bar{S}_B$ is the Stokes vector of the background condition which is a function of the receiver wavelength window, $\lambda_{R_x}$. The terms of the equation are organized by their functional order because matrix operations do not generally commute. The observation matrix is also included because only intensity can be measured directly with the

full Stokes vector determined through measurement with particular configurations of the analyzer (Hayman and Thayer, 2012). Finally, the standard assumptions used to derive the lidar equation including independent and single scattering are also used here, though they are not strictly required. For more information on the SVLE and its derivation, the reader is referred to Hayman and Thayer (2012).

Elements of $\bar{\bar{F}}$ can be used to describe physical attributes of scatterers beyond simple scattering cross section (Kaul et al.,

2004). The reader is referred to Neely et al. (2013) who describe the polarization retrievals and the physical interpretation of the elements CAPABL measures in detail. This description is extended here by relaxing the assumptions made in that work, namely that the receiver orientations are fixed at $0°$, $45°$, and $90°$ relative to the output linear polarization.

From this general form, Eq. 1, the number of photons to be observed can be derived in each polarization channel, given in Eq. 2, assuming that CAPABL: 1) emits a linear polarization at angle $\phi$ from the tilt axis (yielding the simplification

$\bar{\bar{M}}_{T_x}\left(\bar{k}_i\right)\bar{S}_{T_x} = \begin{bmatrix} 1 & \cos(2\phi) & \sin(2\phi) & 0 \end{bmatrix}^T$), and 2) only measures linear polarizations at angle $\theta$ from the reference transmit polarization, (Neely et al. (2013) Eq. 15 with $A\left(\Gamma_{wp}\right) = \bar{\bar{M}}_{R_x}(2\theta)$). These assumptions have been questioned for some optical systems, e.g. (Di et al., 2016), but have been directly measured for CAPABL with a transmitter polarization purity of 123:1 and a receiver polarization purity of $> 800:1$, resulting in an error of the depolarization ratio no greater than 0.8%. This work uses the definition of the backscattering phase matrix as given by Neely et al. (2013) in their Eq. 5. Note that all

constant terms of Eq. 1 which will cancel when taking signal ratios are lumped into the term $\xi(R)$ such as the measurement solid angle, geometric overlap, range resolution, and atmospheric transmission. The transmission Mueller matrix is thus the identity matrix.

$$N_M(R) = \xi(R)\left[F_{11}(R) + \cos(2\theta)F_{12}(R) + \cos(2\phi)\left(F_{12}(R) + \cos(2\theta)F_{22}(R)\right) + \sin(2\theta)\sin(2\phi)F_{33}(R)\right] \qquad (2)$$

Here the number of measured photons incident upon the photodetector, $N_M(R)$, is a function of transmitted and received

polarization angle $\phi$ and $\theta$, respectively, and is related to the scattering phase matrix terms, $F_{11}(R)$, $F_{12}(R)$, $F_{22}(R)$, and $F_{33}(R)$ which are all functions of range. For CAPABL, $\phi = 45^o$; applying this constraint to Eq. 2 cancels the functional dependency on $F_{22}(R)$. Thus, using three distinct receiver polarization channels: $\theta_1$, $\theta_2$, and $\theta_3$, one can create a set of three simultaneous equations which can be inverted to calculate the Mueller matrix terms of interest which describe backscattering



efficiency, depolarization, and diattenuation (used to determine preferential orientation of scatterers). This set of equations is given in Eq. 3.

$$
\begin{bmatrix} N_1(R) \\ N_2(R) \\ N_3(R) \end{bmatrix} = \xi(R) \begin{bmatrix} 1 & \cos(2\theta_1) & \sin(2\theta_1) \\ 1 & \cos(2\theta_2) & \sin(2\theta_2) \\ 1 & \cos(2\theta_3) & \sin(2\theta_3) \end{bmatrix} \begin{bmatrix} F_{11}(R) \\ F_{12}(R) \\ F_{33}(R) \end{bmatrix} \rightarrow \bar{N} = \bar{\bar{A}}\bar{F}
\tag{3}
$$

The general matrix inverse of $\bar{\bar{A}}$ is given in Eq. 4, which is not a function of range but only receive polarizations. The term
$\zeta$ is introduced in Eq. 5 as a constraint on the validity of the inversion where $\zeta = 0$ results in a degenerate inversion because of receiver polarization selection.

$$
\bar{\bar{A}}^{-1} = \frac{1}{\zeta} \begin{bmatrix} \sin(2\theta_2 - 2\theta_3) & \sin(2\theta_3 - 2\theta_1) & \sin(2\theta_1 - 2\theta_2) \\ \sin(2\theta_3) - \sin(2\theta_2) & \sin(2\theta_1) - \sin(2\theta_3) & \sin(2\theta_2) - \sin(2\theta_1) \\ \cos(2\theta_2) - \cos(2\theta_3) & \cos(2\theta_3) - \cos(2\theta_1) & \cos(2\theta_1) - \cos(2\theta_2) \end{bmatrix}
\tag{4}
$$

$$
\zeta = \cos(2\theta_3)(\sin(2\theta_2) - \sin(2\theta_1)) + \cos(2\theta_1)(\sin(2\theta_3) - \sin(2\theta_2)) + \cos(2\theta_2)(\sin(2\theta_1) - \sin(2\theta_3))
\tag{5}
$$

A general form of depolarization and diattenuation for CAPABL can be expressed in terms of arbitrary observation angles as given in Eq. 6 and Eq. 7, respectively, assuming the condition $\zeta \neq 0$ (for CAPABL $\zeta \approx -2$ calculated from receiver polarizations via atmospheric calibration performed for each measurement). Note that the range dependency of depolarization $(d)$, diattenuation $(D)$, $F_{\#\#}$, and $N_{\#}$ are dropped to simplify the expressions.

$$
d - 1 = \frac{F_{33}}{F_{11}} = \frac{(\cos(2\theta_3) - \cos(2\theta_2))N_1 + (\cos(2\theta_1) - \cos(2\theta_3))N_2 + (\cos(2\theta_2) - \cos(2\theta_1))N_3}{\sin(2\theta_2 - 2\theta_3)N_1 + \sin(2\theta_3 - 2\theta_1)N_2 + \sin(2\theta_1 - 2\theta_2)N_3}
\tag{6}
$$

$$
D = \frac{F_{12}}{F_{11}} = \frac{(\sin(2\theta_3) - \sin(2\theta_2))N_1 + (\sin(2\theta_1) - \sin(2\theta_3))N_2 + (\sin(2\theta_2) - \sin(2\theta_1))N_3}{\sin(2\theta_2 - 2\theta_3)N_1 + \sin(2\theta_3 - 2\theta_1)N_2 + \sin(2\theta_1 - 2\theta_2)N_3}
\tag{7}
$$

The diattenuation equations presented by Neely et al. (2013) in their Eq. 7 and Eq. 20 can be recovered from our Eq. 7 by using $\theta_1 = 45^o$, $\theta_2 = -45^o$, and $\theta_3 = 0^o$ for their Eq. 7 and $\theta_1 = 45^o$, $\theta_2 = -45^o$, and $\theta_3 = \pm 90^o$ for their Eq. 20. The depolarization term presented by Neely et al. (2013) in their Eq. 8 can be recovered with either set of angles from our Eq. 6. For clarity, retrievals performed with equations from Neely et al. (2013) are referred to as traditional or orthogonal as the
polarizations used are orthogonal in Poincare space. The retrievals using Eq. 6 and 7 are referred to as non-orthogonal as they require no such assumption.

## 2.2 Dynamic Range

Polarimetric lidar places stringent requirements on optical detection systems due to the large dynamic range of observed signals. It is important to note that the total dynamic range of observed signals arrises from many terms in the SVLE. This



work will parse two, which will be referred to as range induced dynamic range and detector induced dynamic range. Range induced dynamic range arises from the solid angle term, $A/R^2$, and causes signal strength to vary significantly over the altitude range of interest, especially for tropospheric lidar systems. From the initial signal overlap range, $50\,m$, to the $5\,km$ this term changes by 4 orders of magnitude. This is 2 orders of magnitude greater than the change from $10\,km$ to $100\,km$.

5       Detector induced dynamic range arises from having different signal intensities at the same altitude caused by polarization selection, mathematically defined by Eq. 2 by picking different values of $\theta$. An example of this is the frequently used depolarization ratio which is defined further in Sect. 2.3. If one were to consider the differences in signal strength of a depolarization ratio of 1% vs 100%, there is substantial variations as well. A depolarization ratio measurement of 1% indicates parallel and perpendicular polarization signals vary by 2 orders of magnitude whereas a measurement of 100% indicates the signals are the

same magnitude.

Combining range induced and detector induced dynamic range, to measure the depolarization ratio of 1% from $50\,m$ to $5\,km$, 6 orders of magnitude would be required from the weakest high altitude perpendicular signal to the strongest low altitude parallel signal. Figure 1 illustrates these dynamic range terms for a general set of polarization signals. These data are taken during a relatively clear air period at Summit. This figure shows raw signals observed from CAPABL using arbitrary

units to highlight the dynamic range caused by the measurement of depolarization and by the solid angle term of Eq. 1. In this case, detector induced dynamic range introduces approximately 1.5 orders of magnitude and range approximately 4.5 orders of magnitude. Adding these together, to measure clear air from $50\,m$ to $10\,km$ would require no less than 6 orders of magnitude.

As a result of this large range in signals, the requirement that the receiver is acting linearly, i.e. that there is a linear correspondence between incident intensity observed at the receiver photodetector (the information provided by the SVLE) and the

number of photons counted or voltage measured, is critically important. This assumption is known to be limiting due to the possibility of multiple photons arriving at the detector at the same time (Whiteman et al., 1992; Donovan et al., 1993; Liu et al., 2009; Newsom et al., 2009). Pulse pileup, referred to throughout this work as saturation, results in the under-representation of signal intensities in some portions of the observed lidar profile while leaving other portions unaffected. Critically, saturation can affect different polarization channels in an uneven manner and, therefore, directly cause biases in geophysical retrievals

when using signal intensity. In Fig. 1, this is seen clearly as saturation affects the photon counting parallel signal below $6\,km$ but the photon counting perpendicular signal only below $2\,km$.

## 2.3   Depolarization Ratio and Saturation

To fully demonstrate retrieval errors caused by saturation, the traditional linear depolarization ratio, $\delta$, is considered, given in Eq. 8 in its standard form as well as its relation to depolarization $d$ (Schotland et al., 1971; Intrieri et al., 2002; Flynn et al.,

2007; Shupe, 2007; de Boer et al., 2009; Hayman and Thayer, 2009, 2011). It should be noted that depolarization $d$ is a Mueller





matrix element only resulting from the polar decomposition procedure of Hayman and Thayer (2012); depolarization ratio, $\delta$, has been often related to hydrometeor phase but is not an element of the Mueller formalism directly.

$$\delta(R) = \frac{S_{0_\perp}(R)}{S_{0_{||}}(R)} = \frac{d(R)}{2 - d(R)} \qquad (8)$$

In this equation, $\delta(R)$ is the depolarization ratio as a function of range, $S_{0_\perp}$ is the photon arrival rate at the detector surface

in the perpendicular channel as a function of range, and $S_{0_{||}}$ is the photon arrival rate at the detector surface in the parallel channel as as function of range. These are the theoretical arrival rates and not the observed arrival rates. If careful attention is not paid to signal saturation, then the stronger parallel signals may be underestimated causing the depolarization or equivalently the depolarization ratio to be overestimated which can be easily mistaken for geophysical signatures. If, for example, a non-paralyzable system is assumed, corrected for CAPABL's photon counting system based on the analysis provided in Appendix

A, Eq. 8 can be recast as given in Eq. 9 to link the theoretical arrival rates to observed depolarization ratio.

$$\delta_O(R) = \frac{\frac{S_{0_\perp}(R)}{1 + \tau S_{0_\perp}(R)}}{\frac{S_{0_{||}}(R)}{1 + \tau S_{0_{||}}(R)}} = \delta(R) \frac{1 + \tau S_{0_{||}}(R)}{1 + \tau S_{0_\perp}(R)} \qquad (9)$$

Here $\delta_O(R)$ is the observed depolarization ratio and $\tau$ is the photon counting system dead time described by Donovan et al. (1993). CAPABL's photon counting acquisition is best modeled as a non-paralyzable system with a dead time of approximately $6\,ns$, therefore, depolarization ratio values observed from parallel channel count rates exceeding 1 to $10\,MHz$ are noticeably

biased high. Qualitatively, this effect can be seen in Fig. 1 where the signal induced dynamic range is virtually constant for analog detection but a strong function of height for photon counting detection.

     Depolarization ratio error is shown quantitatively in Fig. 2 for many possible ways of measuring depolarization. A similar procedure is performed as in Fig. 1 where photon count rates are modeled from the SVLE and then used in the retrievals given by Eq. 9 but starting with Eq. 6 and applying a combination of receiver polarization angles into the depolarization ratio

calculation. This is done for 6 sets of polarization angles, roughly equivalent to those measured by CAPABL, to demonstrate the biases inherent in the possible depolarization measurements. The traditional way of measuring depolarization requires parallel and perpendicular signals which maximizes the detector induced dynamic range, given in panel (a) of Fig. 2. Panels (b) through (f) show possible alternatives which either show less sensitivity to saturation or more uniform sensitivity to saturation. Using the threshold of $\delta = 0.11$ defined by Intrieri et al. (2002); Shupe (2007), these biases can, at high count rates, exceed the limit

set between liquid and ice making it impossible to observe liquid water even if the true depolarization is smaller than the set threshold, i.e. $\delta_O(R) > 0.11$ when $\delta(R) \leq 0.11$. It should be observed that the effect of saturation shown in panel (a) is neither uniform with count rate or true depolarization ratio nor is it negligibly small relative to the limit set between liquid and ice. The alternatives in panels (b) through (f) offer more uniformity and or reduced bias.

     In the polar regions, given that most liquid clouds are relatively low-lying, optically thick, and occur in all seasons, saturation

will affect signal levels frequently (Intrieri et al., 2002; Turner, 2005; Shupe et al., 2006; de Boer et al., 2009; Shupe et al.,





2011; Shupe, 2011; Shupe et al., 2013; Cesana et al., 2012). This is true regardless of the counting method employed (note the methods are described in more detail in Sect. 3). For photon counting saturation will become prominent and for analog detection pulse heights can exceed analog to digital converter bounds (clipping). This saturation will directly bias depolarization values, which will ultimately cause the misrepresentation of liquid clouds as ice clouds and clipping will result in areas that

are unobservable. The results of this paper quantify the extent of the impact for CAPABL for a 4 month period from July 2015 to October 2015.

## 3  The Clouds Aerosols Polarization and Backscatter Lidar

The CAPABL system has been deployed to Summit, Greenland within the ICECAPS sensor suite since 2010 (Shupe et al., 2013; Neely et al., 2013). The basic operation and measurement principle is well described by Neely et al. (2013) based on

the polarization theory developed by Hayman and Thayer (2012). While the basic measurement principle and resulting raw data products have remained constant since installation, several hardware modifications have improved the system's overall observational capacity. These hardware modifications were completed in June 2015. These modifications are briefly described with an emphasis on how they allow the CAPABL system to better observe clouds via enhancement of counting system dynamic range.

After several years of data collection, the original Nd:YLF laser described by Neely et al. (2013) was replaced by a more powerful Nd:YAG laser. This changed the laser wavelength from $523\,nm$ to $532\,nm$. The optical components were accordingly changed. In addition, the telescope was replaced by a smaller Schmidt Cassegrain telescope to allow the system to be more easily tilted; the current tilt angle is $32°$ from vertical. The photo multiplier tube (PMT) was upgraded from the original PMT, a Thorn EMI 9863B/100, to a Hamamatsu R7400U-03. The current system specifications are given in Table 1, which can be

compared to Table 1 from Neely et al. (2013) for reference.

The major change was an upgrade of the receiver counting system from a purely photon counting system to a combined analog and photon counting system. Photon counting systems are capable of measuring weak light signals, while analog systems sacrifice sensitivity to measure stronger signals. In photon counting, detector signals are discriminated with a fixed voltage threshold. This threshold is set to remove much of the electrical noise resulting from using single-photon, high-gain

PMTs. When a voltage signal is observed in excess of the threshold, a photo-electron is counted and its time of flight is assigned to a particular time bin. The intensity is presumed to be linearly related to the total number of counts in that bin over some integration period. Error can arise with this technique, however, if photons arrive at the counting system in close succession. It is possible that pulses can pileup in such a way that two or more pulses either overlap in time or pass through the system faster than the counting system can reset itself. In either case, the intensity observed by the optical system is not

linearly proportional to the number of photo-electrons counted because some photo-electrons have not been counted. In analog detection, the discrimination threshold is removed and the voltage produced by the detector is passed through an analog to digital converter with its amplitude providing the relative intensity of the collected backscattered signal. This method requires much higher signal-to-noise ratio than photon counting because of the lack of noise mitigating discriminators. By using a



counting system that combines photon counting and analog detection, saturation is mitigated for high count rates using analog detection, approximately $> 10\,MHz$, while maintaining sensitivity to low count rates, approximately $< 1\,MHz$, using photon counting detection. More about this counting system can be found in Newsom et al. (2009).

## 4 Data Analysis and Cloud Phase Identification

5   Lidar observations are often used to classify clouds using direct backscattered intensity estimates and depolarization signatures (Schotland et al., 1971; Thomas et al., 1990; Sassen, 1991; Goldsmith et al., 1998; Shupe et al., 2008; Winker et al., 2009; Yoshida et al., 2010). The backscattered intensity can be related to the volume backscatter coefficient of the scatters and relates to the scatterers' number density and differential backscatter cross section. The depolarization ratio identifies the aspherical nature of the scatterers enabling ice to be distinguished from liquid. Specifically, liquid clouds have been identified by many studies with low depolarization ratios, $\delta(z) < \sim 0.3\,to\,0.11$, with ice being the complement (Intrieri et al., 2002; Shupe, 2007; de Boer et al., 2009). However, given that the observed depolarization ratio can be biased high based on system count rate using photon counting methods (shown in Fig. 2), low-level liquid-only or mixed-phase clouds with count rates in excess of $\sim 10\,[MHz]$ can appear to contain more ice than they actually do. As polar liquid-only and mixed-phase clouds occur predominantly low in the atmosphere, as observed by Intrieri et al. (2002) at SHEBA, Shupe et al. (2011) at Eureka, and Shupe et al. (2013) at Summit, this indicates a potential bias misidentifying liquid clouds as ice clouds which is not physical.

### 4.1 Definition of Data Masks

Data masks are calculated taking advantage of CAPABL's variety of polarization signal measurements. There are several levels of processing and filtering to ensure data quality. These are implemented in an automatic algorithm. This section will describe the filtering steps. The steps are given in Table 2 and described here in order.

20   CAPABL makes observations with 5 second resolution per polarization angle and scans through 4 polarization angles before returning to the original polarization, taking a total of 20 seconds before returning to the first polarization angle. These polarizations are all linear and were oriented parallel to the outgoing polarization, $0°$, (referred to as par), perpendicular to the outgoing polarization, $90°$ (referred to as perp), approximately $45°$ between parallel and perpendicular polarization (referred to as 3rd channel), and approximately $110°$ from parallel (or $20°$ from perpendicular) polarization (referred to as 4th channel). The outgoing polarization is $45°$ rotated from the tilt axis. These scans are parsed by like polarizations and time integrated to 20 seconds per polarization and spatially integrated to the resolution of 30 meters. Saturation corrections are applied per the method described in Appendix A and by Whiteman (2003). It is important to note that the variance of saturation-corrected photon counting is not simply the variance from Poisson statistics, but when saturation correcting, the error introduced by an inexact model fit is also included which increases the variance; this is taken into account for all error analyses and is described in Appendix A. All data is then background subtracted and subject to an SNR filter. The filter bounds are as follows: photon counting data with less than one photon count per bin after background subtraction and analog voltages less than 1 mv per bin after background subtraction (SNR ratio of approximately -5 dB) are removed. This background subtracted and SNR fil-



tered data is then passed to a speckle filter which interrogated a 5 by 5 pixel region around all observations. Measurements where more than 75% of the surrounding data is removed by the SNR filter are also removed. This yields three sets of quality controlled data referred to as Analog (A), Photon Counting (PC), and Saturation Corrected Photon Counting (SCPC).

Polarization properties are then calculated for each A, PC, and SCPC dataset by using the procedure describe in Neely et al. (2013). One deviation from the analysis presented in Neely et al. (2013) used here is the removal of the feedback loop for the 3rd and 4th channels; instead an atmospheric calibration range is used in post processing, which performs the same function as the feedback loop on a measurement by measurement basis. The original feedback loop described by Neely et al. (2013) was designed to accommodate slight retardance changes in liquid crystal variable retarder (LCVR) as a function of ambient temperature. However, in rapidly changing atmospheric scenes, the original feedback loop, designed to eliminate slow systematic effects, was observed to become unstable based on fast atmospheric effects. Using post processing calibration removes the instability by calculating LCVR retardance for each measurement independently. This has been observed to be more stable than the original feedback loop especially in quickly changing cloud scenes and when clouds occupy the pre-determined calibration altitude. This stability has been especially noted when observing low-lying, liquid-only or mixed-phase clouds because of the rapidly changing scene and flexibility of altering the calibration altitude to avoid cloud scenes in post processing.

Depolarization, depolarization ratio, and diattenuation as well as their error estimates are calculated using the standard orthogonal polarization approach presented by Neely et al. (2013), and also using the non-orthogonal approach described above. The orthogonal approach uses all the same steps as the original presentation but with the following exception. Instead of assuming the observations are made at exactly 1) parallel, $0°$, 2) perpendicular, $90°$, and 3) $45°$, the angle of the third channel is carried through the analysis as a variable and the retrieved angle from atmospheric calibration is used. For the depolarization retrieval in areas that lack oriented scatterers, the depolarization can be calculated with any set of measurements of the 6 presented in Figure 2, but for this analysis the strongest 2 signals (par and 3rd channels) were used to demonstrate the range enhancement possible. Orientation is identified by non-zero diattenuation, $D$, for those pixels identified as ice. Diattenuation is calculated in two ways, 1) using par, perp, and the 3rd channel referred to as $D_1$ and 2) using par, perp, and the 4th channel referred to as $D_2$. These channels are chosen because of their opposite sensitivity to saturation for the PC retrievals. By multiplying the two measurements together, negative values indicate $D_1$ and $D_2$ are tending in opposite directions indicating a saturation event. Conversely, positive values of $D_1 D_2$ indicate the two measurements are tending together and that the non-zero diattenuation is physical.

Data is removed outside of the allowable ranges: $0 \leq d \leq 1$, $0 \leq \sigma_d \leq 0.4$, $-1 \leq D \leq 1$, and $0 \leq \sigma_D \leq 0.2$, as these represent non-physical conditions. Note that the error analysis procedure for PC described by Neely et al. (2013) assumes Poisson statistics where the data is assumed shot noise limited. The same procedure for PC is carried through the analysis shown here. The analog signal is not governed by Poisson statistics however. The analog uses the variance of the background voltages for its error estimates. Additionally, as mentioned above, the variance for SCPC is modified to reflect the correction procedure and the variance introduced via inexact model fitting. Finally the backscattering ratio is calculated using temperature and pres-



sure information collected from the ICECAPS twice daily radiosonde program, interpolating between launches, and using the inversion technique of Klett (1981).

Using all of this information, the masking of data is performed in the following manner. Clear air is found as any time and altitude bin, referred to here as a pixel, with a backscattering ratio less than 26. Sub-visible clouds and aerosols are any pixel

with a backscattering ratio between 26 and 50. Clouds are tagged as pixels with backscattering ratio greater than 50. Within cloud pixels, the depolarization ratio threshold, originally defined by Intrieri et al. (2002) of $\delta \geq 0.11$ was used to define ice and $\delta < 0.11$ as water. As the most common aerosol at Summit is ice, any pixels tagged as aerosol that displays a depolarization $\delta \geq 0.11$ is reset as ice. Finally, preferentially oriented ice crystals are identified by $D_1 D_2 > 0.01$ with $\sigma_{D_1}, \sigma_{D_2} \leq 0.05$.

### 4.2   Automatic Algorithm Bounds

The thresholds set for the automated classification algorithm are important to the interpretation of the results of this work. Depolarization and diattenuation are both elements of Mueller matrices, which are defined to have absolute values less than or equal to unity. Values outside this are non-physical. The values on the depolarization and diattenuation error bounds are limited mostly by background count rate, which is tuned via receiver hardware. A receiver neutral density filter lowers both the signal count rates and atmospheric background count rate by a factor of 1000, which brings the signal rates into the desired

dynamic range of the counting system and makes the depolarization and diattenuation error values limited only by shot noise. The filters, which remove data points based on depolarization and diattenuation and their errors, remove less than 3% of all data values. For context, background and speckle filters remove approximately 60% and 23%, respectively, of all data points (data is collected to 35 km). Further, the percent of values removed by the error filters is observed to be fairly insensitive to value changes. This all considered, the bounds for error are set very wide by default.

The setting of the backscatter ratio bounds is more subjective however. As there is no true molecular measurement at Summit (for example provided by a Raman lidar or high spectral resolution lidar), the Klett inversion was used. Curry et al. (1996) note that clear air is uncommon in the Arctic. It has been the authors' experience that even the clearest days at Summit still have some amount of ice in the sky. The clearest day observed within May and June 2015 is used as a baseline to set the clear air threshold. The lowest possible measurements of backscattering ratio with acceptable SNR are in the single digits. Likewise,

the threshold limits between aerosol or sub-visible clouds and clouds were set using an all sky camera. The thinnest visible cloud layer observed during the same time period was used to separate the aerosol or sub-visible clouds and cloud masks.

The threshold between liquid and ice, $\delta = 0.11$, is taken from literature related to the Depolarization and Backscatter Unattended Lidar (DABUL), which was the predecessor to CAPABL, and not changed for this analysis. The same results related to saturation causing anomalously high depolarization ratio values causing a preference for ice over liquid in cloud observations

is observed with lower limits of depolarization ratio. A depolarization ratio split of $\delta = 0.11$ is the most conservative case of this preference based on the literature values between $0.03 \leq \delta \leq 0.11$ and is thus chosen. Lowering the threshold further virtually guarantees no liquid water observations in the PC and SCPC channels based on the information presented in Fig. 2.



### 4.3 Algorithm Example

An example of this data masking procedure is given in Fig. 3 for A detection and Fig. 4 for PC detection. This day is chosen because it contains both single level and two level mixed-phase cloud systems as well as high ice clouds. In comparing these two figures in the first 12 hours of the day, the mixed-phase cloud layer at approximately $1.5\ km$ altitude has been identified

with substantially more liquid pixels when classified using A detection than using PC detection. Furthermore, there are two smaller mixed-phase cloud layers that exist below $1\ km$ between 3 and 5 UTC and 8 to 11 UTC identified by analog detection, which are interpreted as purely ice when classified with PC observations. This interpretation error by PC observations is directly linked to high count rates causing saturation, which increases the observed depolarization ratio beyond the liquid ice threshold.

One other prominent feature observed especially in the analog signals (Fig. 3), and to a lesser extent in the PC signals

(Fig. 4) is multiple scattering. Multiple scattering is known to cause depolarization even within liquid-only clouds due to the possibility of multiple photon paths besides the assumed single backscattering approximation (Eloranta, 1998). When single scattering events are assumed, multiple scattering will produce higher depolarization ratio values that may be classified as ice when, in fact, the scattering volume may contain optically dense concentrations of liquid scatterers. This ultimately makes the tops of some water clouds appear like ice, which is clearly observed from 1 to 8 UTC. There are many techniques to deal with

multiple scattering including multiple field of view lidar systems or post processing tools like those used by Shupe (2007), which reclassify shallow ice layers identified at the top of mixed-phase or liquid-only layers as mixed phase or liquid. For this analysis, multiple scattering clearly skews some of the interpretations towards ice but as the signals from A, PC, and SCPC are all subject to the exact same detector signals, the effect is consistent across all 3 data sets. This results in a constant bias for all three detection methods but as the purpose of this paper is to examine differences between the data sets, multiple scattering is

recognized for future work but not implemented in the masking scheme.

### 5   Observed Cloud Properties

Using the cloud and phase masks described above, monthly statistics are compiled. A single month example is given then multiple months are summarized. Pixels are separated by cloud phase and clear air. Pixels are integrated over the month-long period for each altitude and time bin. These altitude profiles are presented in Fig. 5 for July 2015, the first month of data

available since the hardware updates described in Sect. 3. There is general disagreement in liquid identifications despite the data coming from the same photodetector. The only difference is the method used to handle the electrical signals within the lidar receiver and it is exactly this choice that affects geophysical interpretation. The channel sensitivity can be seen in the ice panel where all lines trend together and with similar slope; analog detection is less sensitive than PC resulting in an offset of the profile values. Profiles for non-orthogonal and orthogonal data as well as PC and SCPC overlap well for clear air above

approximately 3km. Saturation correction at low count rates is akin to multiplying by unity thus resulting in overlapping results between PC and SCPC.

There is a dramatic underestimate of liquid water by CAPABL's PC acquisition, which worsens with decreasing altitude, shown in Fig. 5. At 1000 meters, PC and A differ by 94% (PC observes 34 pixels of water over the month and analog observes





544). This difference is attributed to liquid pixels being mistaken for ice due to saturation in the PC par channel causing erroneously high depolarization ratio values. Below 2.5 km orthogonal and non-orthogonal results are nearly identical but above that altitude non-orthogonal polarization retrievals see more clear air and ice and have higher rates of effective sampling due to the stronger signals used. Recall the 3rd channel measurement is used in the non-orthogonal calculation, which is

stronger than perp measurements and reduces the detector-induced dynamic range between the measurement and the par measurement allowing for greater range-induced dynamic range and, by extension, signal range.

To demonstrate that July data is not anomalous, four months of available data from July 2, 2015 to October 31, 2015 are presented in Fig. 6. Over this time, the CAPABL system had an uptime of $> 99\%$ (this equates to approximately 5 minutes of missed data per day, which occurs at midnight UTC each day to perform system diagnostics and housekeeping). This figure

illustrates any altitude biases in classifying the type of cloud or clear air while Fig. 5 illustrated the occurrence frequencies with height for each identifier. These data are compiled into box-and-whisker plots based on the profiles calculated for each month similar to those presented in Fig. 5. The median altitude of all pixels for each identifier: ice, liquid, and clear, is given as a line through the center of the box, which is completed by the 25th and 75 percentile of all monthly data. The whiskers extend to the 5th and 95th percentiles. The other data values are considered outliers.

Figure 6 indicates 3 prominent features. First, the median altitude of liquid pixels is not constant between A, PC, and SCPC. There is a clear 1 to 2 km offset in the medians between analog and photon counting (1.72 km, 1.43 km, 0.75 km, and 0.91 km offsets for July, August, September, and October, respectively). This offset in mean pixel height indicates that low-level, liquid clouds are often misclassified by the PC channel as indicated by Fig. 3 and Fig. 4, clearly demonstrating that saturation can change the geophysical interpretation of the polarimetric lidar signals and must be considered when detector-induced dynamic

range is high, for CAPABL this occurs approximately when $\delta \leq 0.1$. The second feature is seen in the clear sky data where there is increased sensitivity of the PC channel over the analog channel and increased sensitivity of the non-orthogonal polarization retrievals over the orthogonal versions, as noted for the July histogram. This increased sensitivity is seen by the increase in whisker range of approximately 1 km (0.96 km, 0.70 km, 0.34 km, and 0.55 km for July, August, September, and October for SCPC to the 95th percentile, respectively, or 1.17 km, 1.12 km, 0.99 km, and 0.83 km to the inner fence) indicating the presence

of more high altitude clear air pixels that pass the quality control standards specified in Table 2. As a result of the increased sensitivity, the median of the data shifts upwards as well (0.29 km, 0.29 km, 0.36 km, and 0.31 km for July, August, September, and October for SCPC, respectively). The final feature is the relative consistency of the occurrence of ice for all methods. The median altitude of the data shifts slightly upwards again due to increased sensitivity between analog and photon counting (0.05 km, 0.23 km, 0.36 km, and 0.23 km for July, August, September, and October for SCPC and analog, respectively) but the boxes

cover similar altitude ranges, especially for July. Comparing the whiskers for the non orthogonal and orthogonal polarization retrievals within a month indicates that the increased sensitivity gained by using non orthogonal polarization retrievals does not change the geophysical interpretation of the data when saturation is of little concern (shifts of 0.26 km, 0.08 km, 0.21 km, and 0.10 km for July, August, September, and October for analog to the 95th percentile, respectively, or 0.18 km, 0.13 km, 0.21 km, and 0.18 km to the inner fence are observed), i.e. when signals are of similar strength or when signal rates are less than

approximately $1\,MHz$.





## 6    Discussion

### 6.1    Selecting Dynamic Range

CAPABL is a single detector lidar system with signal sensitivities to four, sequentially scanned, polarization-planes. The polarization of the receiver is set using a LCVR in combination with a quarter wave plate to create a variable rotator. This design has proven robust and simple which is especially useful as the investigators have little access to the instrument throughout the year. It does however require that the total signal dynamic range be set once a year. The range is set using a neutral density filter in the receiver to attenuate the signal. A neutral density of 3 is used to insure signals from clear air scattering do not exceed 10 MHz. The receiver design is similar to a polarization sensitive micro-pulse lidar (MPL), described by Flynn et al. (2007), with a single fixed dynamic range but operating with linear polarizations only.

The LCVR voltages for CAPABL's 4 planes of linear polarization are set to provide evenly spaced signal intensities from clear air scatterers, not to the exact angle relative to the transmit polarization. The signal increments were made such that the polarizations for the 3rd and 4th channels were set to more evenly cover the signal range between the par and perp polarizations. In clear air calibration tests, perpendicular signals were approximately 2 orders of magnitude less than parallel (detector induced dynamic range). The 3rd channel was set to be approximately a factor of 3 less than the parallel channel and the 4th channel approximately a factor of 8 less. This allows the detector dynamic range of interest to be selected based on the chosen polarization channel. For example, in a low-level liquid cloud, the three weakest signals can be used. For weaker scattering objects like clear air, aerosols, sub-visible cirrus clouds, or objects further in range, the three strongest signals are used. In clear air with a depolarization ratio of approximately 1%, the difference in signal strength, detector induced dynamic range, between par and perp is 2 orders of magnitude. Assuming 5 total orders of magnitude of linear operation for the specified counting system and a fixed preset dynamic range, this low depolarization ratio only allows the system to have a range induced dynamic range of approximately 3 orders of magnitude. From 50 meters to 5 km, a range induced dynamic range of 4 orders of magnitude is required. Using par and perp channels makes this untenable but using non-orthogonal polarization channels reduces the detector induced dynamic range by approximately a factor of 10 to 12 allowing for an extra order of magnitude fluctuation with range.

Using non-orthogonal polarization retrievals has extended the altitude range of CAPABL by as much as 1 to 1.5 km. Evidence of this increase can be seen in Fig. 5 where the total percent of pixels that pass the quality control process increase using non-orthogonal polarization retrievals. At Summit, where clouds rarely occur above 8 km AGL, this equates to a 12.5% to 18.75% enhancement in effective sampling of the desired altitude range.

### 6.2    Optimum Combination of Signals

The results of this work highlight the differences in signal dynamic range that propagate through the provided analysis altering the physical interpretation of the measurements made. While combining the measurements into the optimum combination of signals is beyond the scope of this work, it is useful to broadly understand the way to combine all the different signal approaches to best utilize the available data to extend the work started with CAPABL to different lidar systems. Two approaches are



noteworthy. The first is to combine A and PC signals at the raw signal level to create a "glued" profile and to use the resulting profile in all retrievals. The second approach is to avoid "gluing" the profile together and to combine data masks after processing based on raw signal strengths and error estimates. The latter method is preferred in this analysis for a few primary reasons. First, it is unclear what quantity of error is introduced in the "gluing" procedure as it appears to be sensitive to background

or noise sources and time (Newsom et al., 2009). Further, the "gluing" procedure produces slightly different results based on the exact methodology. Implicit in the gluing procedure is the assumption that both photon counting and analog methods show linearity over some overlap signal strength. Therefore, there is a dynamic range where analog is clearly preferred and one where photon counting is clearly preferred. The size of this overlap region is not well characterized but it is assumed to exist. Second, it is unclear what error would be introduced, if any, in using a section of a "glued" profile that results from analog

detection and comparing it to a section resulting from photon counting. This could occur, for example, in a low-level cloud with low depolarization ratio yielding a weak perp signal and a strong par signal. Finally, it is clear when combining results from the second method (calculating separate masks) that the best signal to use is one that uses valid signals with the lowest error estimate. The procedure described in Sect. 4 defines valid signals, and the error estimates without "gluing" are much simpler to define and understand.

The method suggested by analysis of CAPABL's data is to first process the data via orthogonal and non-orthogonal methods. The orthogonal is preferred where all signals are within the counting system's linear range because only one polarization angle (either the 3rd or 4th channel) needs to be retrieved. Two angles are known and one is used introducing only 4 error sources (shot noise on 3 channels and the error of the retrieved angle). If one signal is either too strong or too weak, the non-orthogonal retrievals can increase the valid retrieval range by extending range-induced dynamic range by trading detector-induced dynamic

range. There are 5 error sources (shot noise on 3 channels and the error of the two retrieved angles) and generally higher error estimates. Finally when adding in analog and photon counting, photon counting is preferred for low signal strengths but saturation correction typically has higher error than simply using analog detection. Putting all these findings together, in terms of range, non-orthogonal polarization retrievals with the weakest signals are used for the near range, orthogonal for mid ranges, and non-orthogonal polarization retrievals with the strongest three signals for far ranges are suggested by this

analysis. In terms of signal strengths, the use of photon counting and analog signals are suggested while avoiding the saturation correction procedure altogether based on the additional error introduced with the uncertain model fit and the assumption stated above that there is a range of signal strengths where both photon counting and analog are valid yielding no requirement to extend the valid range of photon counting into the analog range.

### 6.3 Radiative Implications of Cloud Phase Misidentification

Cloud phase is shown to be an important driver of the radiation budget. For a constant amount of water, the liquid phase affects the radiative budget more strongly because it tends to form many small droplets with larger surface area where ice tends to form larger crystals. As such, the optical depth and longwave emission of liquid is greater than of ice. At Summit, Miller et al. (2015) show that cloud radiative forcing is driven primarily by the annual variability in liquid bearing clouds. It is therefore important to assess the CAPABL data in terms of the fractional occurrence of liquid, ice, and clear air states.





Figure 7 gives the fractional occurrence of pixel types in the column above CAPABL. Profiles are defined from the ground to 15 km for a given time by the radiatively dominant pixel type, providing a vertically integrated snapshot of the profile above Summit. For example, if a profile contains liquid water, ice, aerosol, and clear air, it is defined as liquid. Ice is defined as any column with ice pixels in it but that lacks liquid. Sub-visible columns must contain that pixel type without ice or liquid cloud

pixels present. Clear air columns must contain nothing but clear air. In this way, one can convert the pixel number defined in Fig. 6 to a cloud fraction proxy.

An examination of Fig. 7 suggests that all 6 processing types: A, PC, SCPC for both orthogonal and non-orthogonal, yield similar fractional occurrence of clear air pixels (differences of 8%, 2%, 1%, and 3% for July, August, September, and October, respectively). Figure 7 also shows a distinct difference in liquid and ice column identification (differences of 31%, 35%, 26%

and 24% for July, August, September, and October, respectively, for A and PC). This difference in liquid water cloud fraction can be used to approximate an error in cloud radiative forcing using the results from Fig. 7 from Miller et al. (2015). Using an approximate difference of 30%, this time period of fractional occurrence of liquid clouds equates to an error in longwave cloud radiative forcing of approximately $10[W/m^2]$. Miller et al. (2015) finds an average of $33[W/m^2]$ for cloud radiative forcing at Summit suggesting that using uncorrected CAPABL data to infer radiative impacts could under-represent forcing by as much

as one third.

### 6.4   Recognized Future Work

For certain designs, consideration of the signal dynamic range may be as important as the selection of polarization planes. The same problems related to dynamic range that are demonstrated for CAPABL could exist in a one detector design, like the polarization sensitive MPL, because the perpendicular and circular polarizations can still vary by as much as two orders of

magnitude in detector induced dynamic range for very low depolarization ratio targets like liquid-only clouds, mixed-phase clouds, and clear air. More work is suggested to determine the possible errors in single detector designs as they are often designed to use orthogonal polarizations, which have vary different dynamic ranges.

One of the major topics to discuss is the handling of multiple scattering. Multiple scattering tends to increase signal strength but is important primarily within regions of high optical thickness. Even with scatterers that are purely spherical, multiple

scattering can cause signal depolarization. In the CAPABL data set, this is most noticed in the middle and top of low-level liquid-only and mixed-phase clouds. The focus of this paper has been differences caused by count rate and signal strengths. In regions of multiple scattering, signal count rates tend to be low due to attenuation of signal. There is an increased proportion of signal in the perp polarization even though the overall signal is being attenuated through the region. However, as all signals used come from the same volume and time via the same detector signals, the result is a bias in all channels. There is no bias that

manifests itself in only one channel, indicating that the differences observed between A, PC, and SCPC for both orthogonal and non-orthogonal retrievals must be due to other effects. The effect of multiple scattering is suggested for future work to further refine the measurement capabilities of CAPABL.

One major finding from analysis of 5 years of CAPABL data is the sensitivity of the diattenuation measurement to saturation. One can see clearly in Fig. 4 that the areas identified as having erroneous depolarization measurements have diattenuation





values in excess of 0.4. The combined diattenuation product, described in Sect. 4.1 and Table 2 in step 10, can distinguish between preferential orientation and saturation. This finding was used manually and not operationally within this work but will be included in future retrieval versions to enhance the quality of the retrieval. For systems that have the ability to measure more than 2 independent polarization channels, diattenuation is found to be a sensitive measure of retrieval validity.

## 7    Conclusions

Ground based measurements of cloud properties are critically needed in the polar regions to help improve modeling studies of major weather and climate processes. A particular need is to identify and distinguish cloud liquid water from ice as their roles in the radiative balance of the polar regions are distinctively different. This paper has highlighted the challenges in identifying liquid-only and mixed-phase clouds by polarimetric lidar observations, and the utility of employing multiple polarization planes

to cover the signal dynamic range, or equivalently, cover the diversity of signals introduced by the variety of cloud types and altitudes. The estimate of depolarization ratio, which has been often correlated to cloud phase, is dependent not only on the differential intensity in the polarized signal caused by the observed scattering process but the lidar count rate observed. Lidar count rates are related, as seen via the Stokes vector lidar equation, to cloud optical depth, observation range, and the response of the photodetector and counting system. These additional dependencies cause biases in lidar data as the signal dynamic range

changes significantly causing the strongest polarization signals to saturate, which manifests in non-physical increases in the depolarization ratio. This directly biases the interpretation of observations to preferentially identify ice over liquid water. The predominantly low-lying and optically thick nature of liquid clouds over Summit, Greenland, relative to ice clouds, result in very strong signal diversity that must be treated in order to properly identify and classify cloud types by lidar.

This work has demonstrated three key points. The first point is that cloud phase classification by polarimetric lidar is sensitive

not only to the cloud phase but other cloud properties such as base height (or range) and optical depth, and to lidar design properties such as the power aperture product, field of view, receiver polarization, and detection schemes. The second point is that this associated signal diversity in the lidar observations must be recognized in order to flag conditions unsuitable for determining cloud phase, an inherent problem in two-channel polarization lidars. In high dynamic range targets, like optically thick liquid-only or mixed-phase clouds, such requirements can cause a misrepresentation of liquid clouds as ice clouds. The

final point is that by employing multiple planes of polarization in the lidar receiver, in the case of CAPABL four linear planes, the diversity in backscattered intensity may be handled more judiciously making the characterization of cloud types more accountable. This effectively spreads the required dynamic range of signals among the multiple polarization measurements. Furthermore, the signal dynamic range in each polarization is extended by incorporating both analog and photon counting capabilities. This polarization configuration and signal combination allows the CAPABL system to self analyze limitations in a

channels performance, correct some of the behavior, and optimize the use of the different channels for different cloud scattering conditions.

A case study of 4 months of data from the CAPABL system at Summit, Greenland is shown to demonstrate how signal saturation issues in certain polarization channels can be identified and directly linked to geophysical retrieval errors. For



example, the difference in the estimate of the median height for liquid clouds is shown to differ by as much as 2 km between analog and photon counting detection because of photon-counting detection misidentifying the presence of low-lying liquid clouds as ice. This also yields a difference of approximately 30% in estimates of fractional occurrence of liquid clouds. It is further demonstrated that the sensitivity of a polarization lidar system can be enhanced while simultaneously reducing

the required system dynamic range by using non-orthogonal polarization measurements. The range of effectively sampled atmospheric measurements can be extended by as much as 18%, or equivalently 1.5 km, using non-orthogonal polarization measurement.

While this paper has analyzed the biases introduced into CAPABL's lidar data due to counting non-linearity, future work could certainly improve data quality. As highlighted in Sect. 4 and 6, more work to remove the systematic bias caused by

multiple scattering is needed. This data could also be used in a more sophisticated method by leveraging more co-located measurements such as that presented by Shupe (2007). In particular, CAPABL's observations could be combined with collocated MPL, microwave radiometer, and radar Doppler spectra to more precisely classify cloud pixels as possible mixed-phase clouds, while also addressing the inherent ice bias in all polarized signals due to multiple scattering.

## 8   Data Availability

All data collected by the ICECAPS program is publicly available at: anonymous@ftp1.esrl.noaa.gov/psd3/arctic/summit/.

## 9   Code Availability

The code developed to process the CAPABL data is available by request from the authors.

## Appendix A:  CAPABL's Nonlinear Photon Counting

CAPABL's photon counting system is subject to pulse pileup, as is the case with most photon counting systems. This pileup

results in detector pulses occurring too close in time for the counting system to uniquely identify individual pulses, resulting in systematic underrepresentation of photon count rate. The models introduced to correct this problem are based on the work of Donovan et al. (1993); Whiteman (2003); Liu et al. (2009) using a calibration data set taken during a clean air period at Summit in May 2015. The neutral-density filter was removed from the receiver optical path on a clear air day to increase the observed count rate and also extend the vertical range of calibration data. Data were concatenated based on the work of Newsom et al.

(2009) with the main difference being that profiles were background subtracted before analysis (note that this is the only case in this manuscript where such concatenation is performed). From these data, the analog profile is taken as the ideal count rate. These data are plotted in Fig. 8 with two correction methods fit to the data using a Levenberg-Marquardt nonlinear least squares



solver. These saturation models are given in Eq. A1 and Eq. A2, referred to as non-paralyzable and paralyzable, respectively. The fit parameter for non-paralyzable is the deadtime $\tau_{NP}$ and for paralyzable $\tau_P$.

$$S_{obs} = \frac{S_0}{1 + \tau_{NP}S_0} \tag{A1}$$

$$S_{obs} = S_0 \exp\left(\tau_P S_0\right) \tag{A2}$$

To convert from the observed photon count number to observed photon count rate, a simple linear transformation is given in
Eq. A3 where $N_{obs}$ is the observed photon count number per bin, $S_{obs}$ is the observed photon count rate per shot, $S_{PP}$ is the number of laser shots integrated per profile, and $T_{PB}$ is the two way travel time of light per range bin.

$$N_{obs} = S_{obs} \times S_{PP} \times T_{PB} \tag{A3}$$

Inserting Eq. A3 into Eq. A1 and performing a propagation of error analysis, based on Taylor series expansion for standard error propagation assuming no data covariance, yields the shot noise error for the corrected photon count number per bin.

$$\sigma_N = S_{PP}T_{PB}\sqrt{\frac{N_{obs}^4\sigma_{\tau_{NP}}^2 + S_{PP}^2T_{PB}^2\sigma_{N_{obs}}^2}{\left(S_{PP}T_{PB} - \tau_{NP}N_{obs}\right)^4}} \tag{A4}$$

Equation A4 indicates that the error in corrected photon count rate is a function of the count error $\sigma_{N_{obs}}$ which conform to Poisson statistics and the error in the model fit parameter $\tau_{NP}$. This error is estimated during the fitting procedure using the fit confidence bounds. Note that if and only if $\tau_{NP}$ is exactly zero (i.e. $\tau_{NP} = 0$ and $\sigma_{\tau_{NP}} = 0$) will the counting error be simply $\sigma_{N_{obs}}$.

The calibration data used for this analysis is presented in Fig. 8 where the fitting regions are $0.1\,MHz$ to $500\,MHz$. As each measurement is subject to some measurement error, Poisson counting error for photon counting and electrical noise for the analog detection, this fit was calculated using the signal to noise ratio (SNR) as a data weight such that higher SNR data are given higher weights. The results of this weighted analysis indicate that the dead time is approximately $0.1\,[ns]$ higher than the unweighted analysis which ignores measurement errors in the fit.

*Author contributions.*   R. Stillwell prepared the manuscript with contributions from all co-authors.

*Acknowledgements.*   This material is based upon work supported by the National Science Foundation Graduate Research Fellowship Program under grant No. DGE 1144083 and National Science Foundation grants No. AON 1303864, PLR-1303864, PLR-1303879, PLR-1314156, and ATM-0454999. Ryan Neely is funded by the National Centre for Atmospheric Science. The authors would like to thank the staff and science technicians at Summit as well as Polar Field Services for their support and dedication to help collect and maintain instrumentation.





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





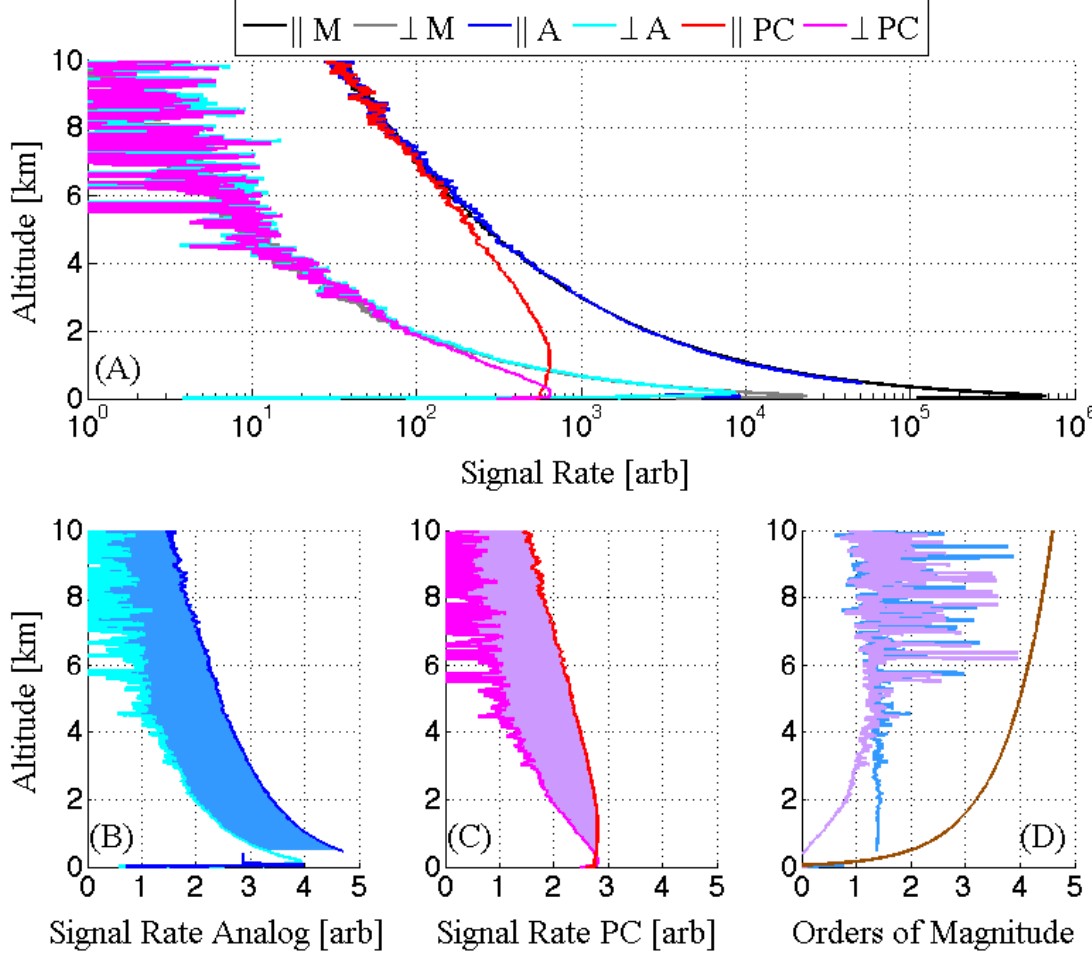

**Figure 1.** Measured and theoretical count profiles for the CAPABL system, panel (A). Measurements taken from May 26, 2015 are shown from the CAPABL system of its parallel ($\phi = 45°$ and $\theta = 45°$) and perpendicular ($\phi = 45°$ and $\theta = -45°$) polarization angles. Eq. 1 is used to model the same assuming a perfect measurement system, labeled M. Analog (A) and photon counting (PC) measurements are included to verify the validity of the modeled results. There is general disagreement of the modeled and photon counting data above signal rates of $10^2 [arb]$ due to saturation and above $10^{4.5} [arb]$ for analog due to pulse heights clipping the analog to digital converter. Analog parallel and perpendicular signals are highlighted in panel (B) with the detector induced dynamic range colored in light blue. Photon counting parallel and perpendicular signals are highlighted in panel (C) with the detector induced dynamic range colored in pink. The detector induced dynamic ranges are shown in panel (D) colored light blue for analog and pink for photon counting with range induced dynamic range, introduced by the $R^2$ term in Eq. 1, referenced to $50 [m]$, the theoretical and observed overlap distance, in brown. For reference, the total advertised linear dynamic range of the counting system used to make this measurement is 5 orders of magnitude.





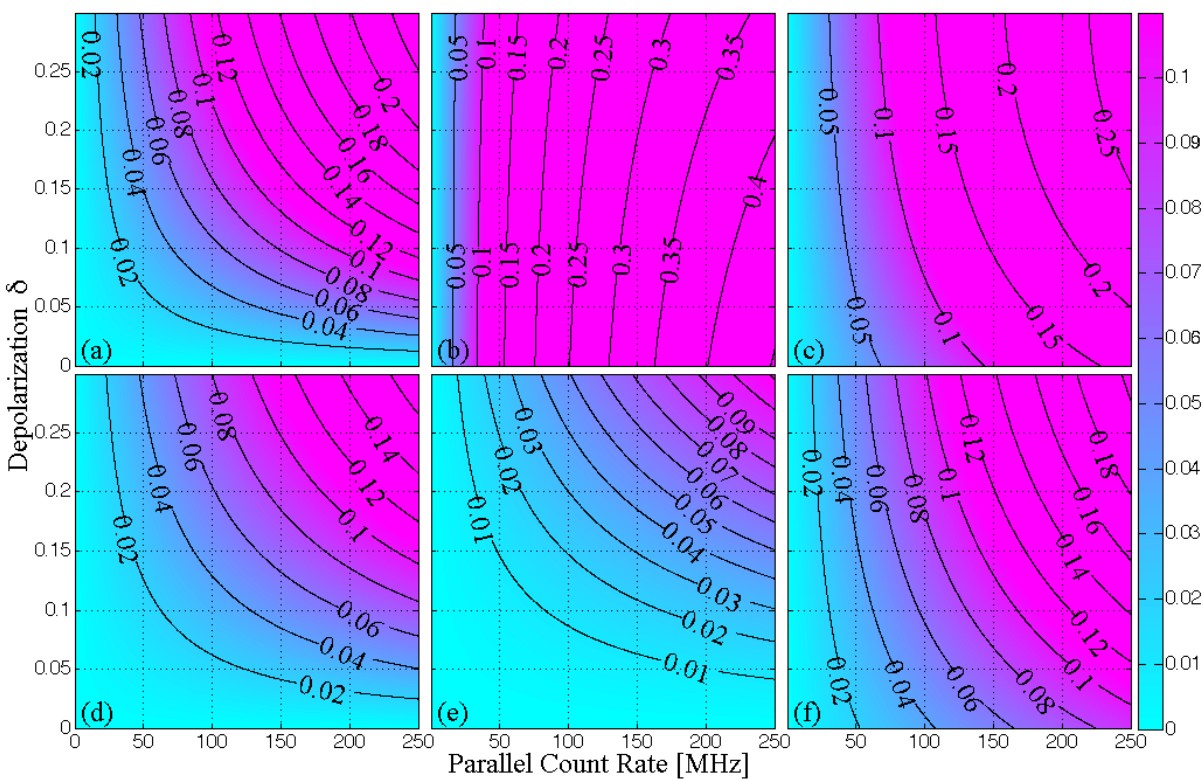

**Figure 2.** Theoretical deviation of the observed depolarization and the true depolarization ($\delta_O - \delta$) as a function of the parallel count rate and the depolarization ratio ($\delta$). The count rates for all CAPABL channel is calculated using Eq. 2 and the depolarization using Eq. 6. Assuming zero diattenuation, only two channels are required for the inversion. The channels used to calculated each contour are: a) $\theta_1 = 0°$ and $\theta_2 = 90°$ (traditional), b) $\theta_1 = 0°$ and $\theta_2 = 45°$, c) $\theta_1 = 0°$ and $\theta_2 = 110°$, d) $\theta_1 = 90°$ and $\theta_2 = 45°$, e) $\theta_1 = 90°$ and $\theta_2 = 110°$, and f) $\theta_1 = 45°$ and $\theta_2 = 110°$. The results presented in this paper as orthogonal come from calculations similar to that given in panel a) and non orthogonal given in panel b). The color bar is scaled to match the adopted thresholds for liquid water, $\delta = 0.11$ as defined by Intrieri et al. (2002); Shupe (2007).





**Figure 3.** Analog data from the CAPABL system for February 29, 2016. Total Backscatter is the summation of background subtracted parallel and perpendicular voltages converted to a virtual count rate (V.C.R.) using a data gluing procedure in MHz. The total backscatter color bar is given from 100 KHz to 250 MHz on a logarithmic scale. Depolarization is calculated as given in Eq. 6. Diattenuation is calculated as given in Eq. 7. Backscatter ratio is calculated by performing a Klett inversion and using ICECAPS radiosonde data (launched at 2400 UTC and 1200 UTC daily) to calculate a molecular extinction component (Klett, 1981). The data mask given is calculated as given by the rules described in Sect. 4. Liq., S.V., and Cl. stand for liquid, sub-visible, and clear, respectively.





**Figure 4.** Photon Counting data from the CAPABL system for February 29, 2016. Total Backscatter is the summation of background subtracted parallel and perpendicular photon counts converted to count rate (C.R.) in MHz. The total backscatter color bar is given from 100 KHz to 250 MHz on a logarithmic scale. Depolarization is calculated as given in Eq. 6. Diattenuation is calculated as given in Eq. 7. Backscatter ratio is calculated by performing a Klett inversion and using ICECAPS radiosonde data (launched at 2400 UTC and 1200 UTC daily) to calculate a molecular extinction component (Klett, 1981). The data mask given is calculated as given by the rules described in Sect. 4. Liq., S.V., and Cl. stand for liquid, sub-visible, and clear, respectively.





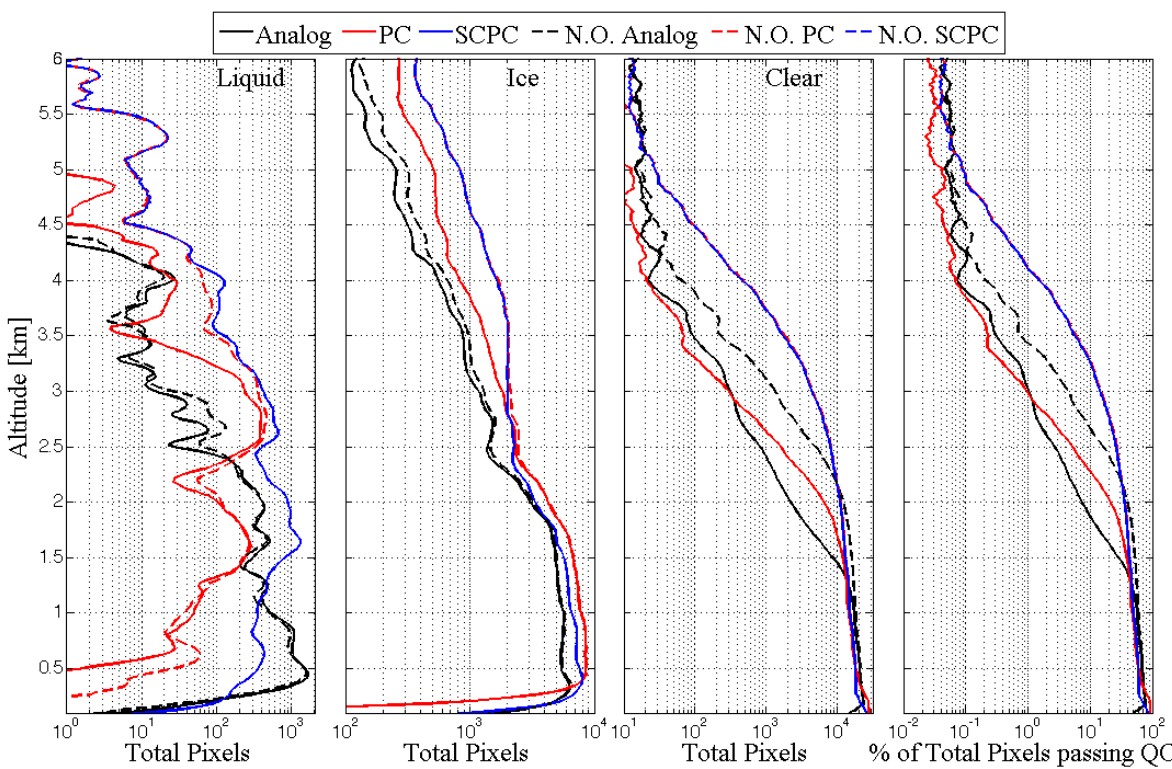

**Figure 5.** Histograms of all the monthly data collected in July, 2015. All pixels observed which pass the criteria described in Table 2 are included. The panels labeled Liquid, Ice, and Clear are summed pixels and the final panel without a labels is the percent of possible pixels observed. The legend descriptor N.O. indicates non-orthogonal calculation of polarization properties and those without indicate standard orthogonal calculation procedures. Note that the sensitivity of the channel is given quantitatively by how often measurements at a given height pass the criteria defined in Table 2. At altitudes above approximately 4 km, most pixels fail the SNR filter except cloud scenes and at altitudes below approximately 200 meters, some data is filtered because the analog detector signals exceed the range set for the analog to digital converter.





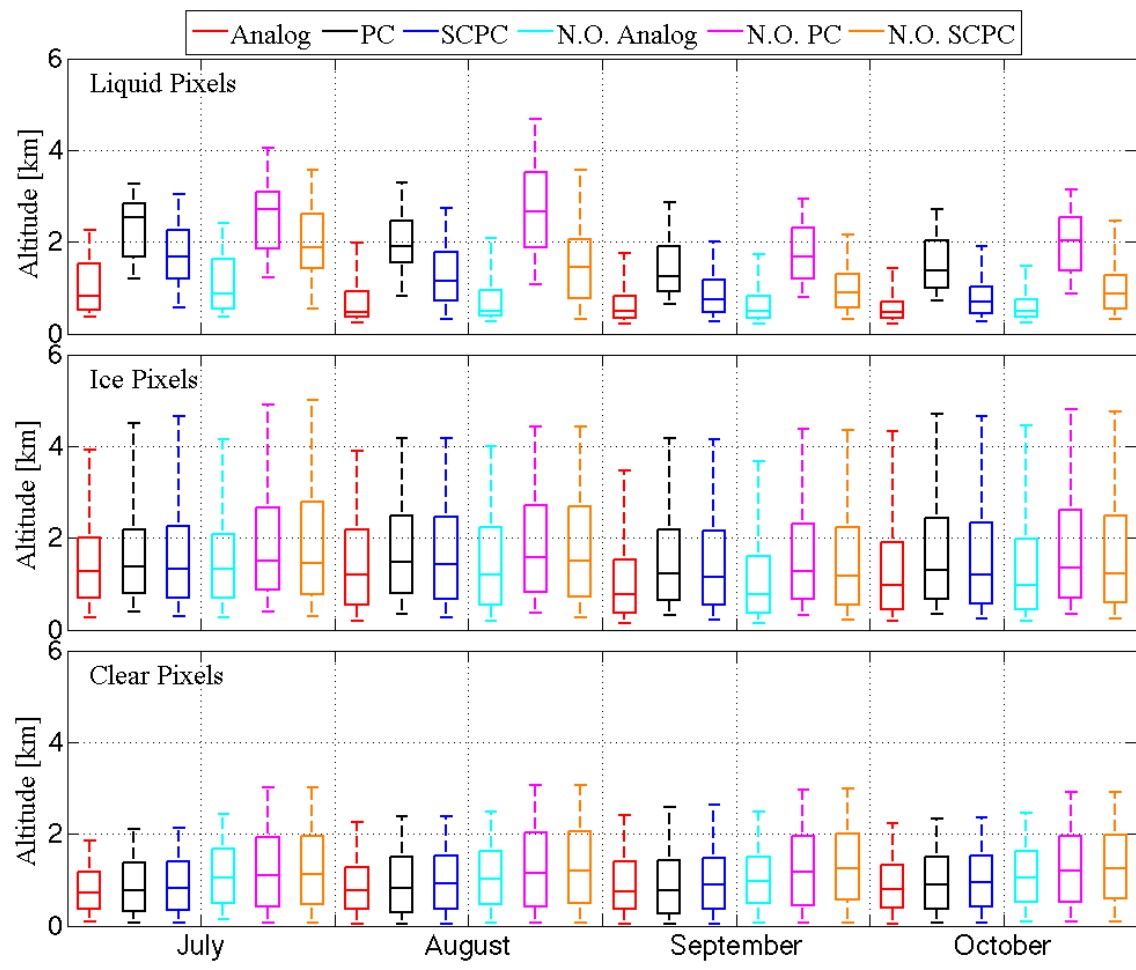

**Figure 6.** 4 months of CAPABL data binned into liquid, ice, or clear air. The median is indicated by a line through the box, the 25th to 75th percentile ranges complete the box and the whiskers extend to the 5th and 95th percentiles. The channel sensitivity can be seen looking at the clear pixels where analog is expected to be less sensitive than PC and orthogonal less sensitive than non-orthogonal. Note also that there is a significant deviation in the median altitude for liquid water observed via PC and via analog detection.




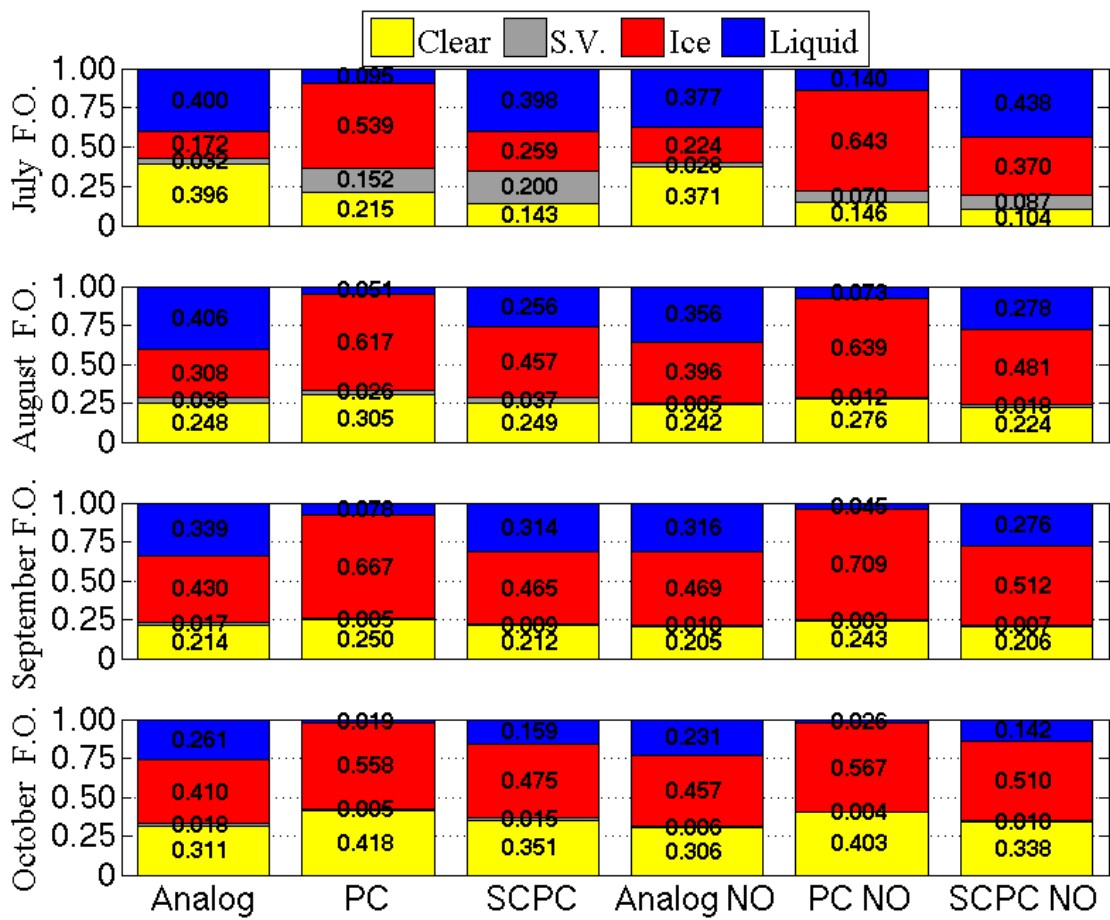

**Figure 7.** Fractional occurrence (F.O.) of each pixel type in the column for each month. To be labeled clear, the column must lack all sub-visible, ice, and water pixels. To be labeled sub-visible, the column must lack ice or water pixels. To be labeled as ice, a column must lack water pixels. If a column contains a water pixel, the column is labeled as liquid. The fractional occurrences are given for each bar rounded to the nearest thousandth.



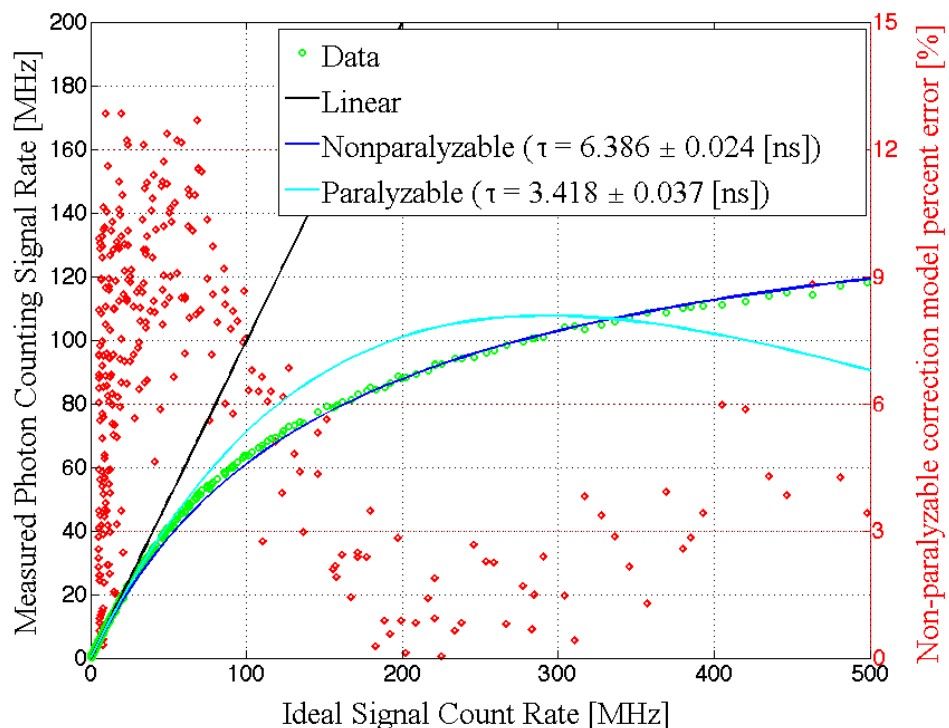

**Figure 8.** Saturation analysis of the CAPABL photon counting channel using the theory developed by Donovan et al. (1993); Whiteman (2003); Liu et al. (2009). The ideal signal count rate is found by normalizing the analog detection channel to the photon counting channel in a region where both are acting linearly which is about 500 kHz count rate. The measured count rate is then taken directly from photon counting measurements. The paralyzable and non-paralyzable models are then fit using a Levenberg-Marquardt weighted non-linear least squares fitting algorithm of the observed calibration data. The 1σ confidence bound is given for each dead time fit parameter. Finally, the percent error of the correction model is given relative to the ideal count rate on the right ordinate as diamonds.



**Table 1.** CAPABL current system specifications. Note that polarization purity and polarization rejection are measured quantities. Polarization purity is measured with a 100,000:1 Glan-Taylor polarizer.

| Transmitter | Receiver | Signal Processing |
|---|---|---|
| Big Sky Laser Ultra flashlamp pumped Nd:YAG | Schmidt Cassegrain Telescope | Combined Analog and Photon Counting acquisition |
| Wavelength: 532.3 nm | Receiver Aperture: 20.8 cm | Data system: |
| Pulse Energy: 60 mJ | Filter Bandwidth: 0.3 nm | Licel Transient Recorder TR20-12 Bit |
| Pulse Rate: 15 Hz | Channels: 1 | Range bin size: 7.5 m |
| Twin Head | Field of View: 1.4 mrad | Integration time: 5 sec |
| Polarization Purity: $> 123 : 1$ | Polarization Rejection: $> 800 : 1$ | PMT: Hamamatsu R7400U-03 |
| | Linear Polarizations Observed: 4 | |



**Table 2.** A summary of the data processing steps taken to create the data masks desired for CAPABL. The processing for each data type: Analog (An), Photon Counting (PC), and Saturation Corrected Photon Counting (SCPC), is constant except where noted. Note that the diattenuation error equation is calculated per standard propagation of error techniques taking a Taylor series expansion of Equation 7.

|     | Processing Step | Channels | Details |
| --- | --- | --- | --- |
| 1) | Time integration | An/PC | To a constant 20 second resolution |
| 2) | Spacial integration | An/PC | To a constant 30 meter resolution |
| 3) | Saturation correction | PC | Creates SCPC level |
| 4) | SNR filter | All | |
| 5) | Speckle filter | All | $5 \times 5$ surrounding box |
|     |                 |     | $> 75\%$ data already removed $=$ bad |
|     |                 |     | $> 25\%$ data available $=$ good |
| 6) | Calculate polarization properties | All | Depolarization and depolarization ratio per Eq. 6 and 8 |
|     |                 |     | Depolarization and depolarization ratio error per error propagation of Eq. 6 and 8 |
|     |                 |     | Diattenuation per Eq. 7 |
|     |                 |     | Diattenuation error per error propagation of Eq. 7 |
|     |                 |     | Backscatter ratio ($R$) per (Klett, 1981; Neely et al., 2013) |
| 7) | Remove non-physical values | All | Values outside $0 \leq \delta \leq 1$ |
|     |                 |     | Values outside $0 \leq \sigma_\delta \leq 0.4$ |
|     |                 |     | Values outside $-1 \leq D \leq 1$ |
|     |                 |     | Values outside $0 \leq \sigma_D \leq 0.2$ |
| 8) | Calculate base mask | All | Clear: $1 \leq R < 26$ |
|     |                 |     | Aerosol: $26 \leq R < 50$ |
|     |                 |     | Cloud: $R \geq 50$ |
| 9) | Calculate phase mask | All | Liquid: cloud pixels with $0 \leq \delta \leq 0.11$ |
|     |                 |     | Ice: cloud pixels with $\delta > 0.11$ |
| 10) | Calculate orientation mask | All | Random: ice with $0 \leq D_1 D_2 \leq 0.01$ |
|     |                 |     | Preferential: ice with $D_1 D_2 \geq 0.01$ and $\sigma_D \leq 0.05$ |