# Peer review of "Low-Level, Liquid-Only and Mixed-Phase Cloud Identification by Polarimetric Lidar"

_Atmospheric Measurement Techniques, 2016_

## Referee Comment (RC1) · Anonymous Referee #3 · 15 Mar 2017

This is a review of the manuscript "Low-Level, Liquid-Only and Mixed-Phase Cloud Identification by Polarimetric Lidar" submitted by Stillwell et al. to AMTD.

The paper explores different methods to measure lidar depolarization using the CA-PABL lidar. Specifically, uncertainties in the determination of thermodynamic phase of clouds in the Arctic caused by depolarization measurement errors are discussed.

This is a highly technical paper. Unfortunately, for a casual lidar data user as myself, the relevance of the work is difficult to assess. The paper shows differences in measured depolarization and cloud classifications when using different measurements strategies, but it does not provide an objective validation about which strategy is more accurate. It also does not adequately discuss the implication that this work might have on previous statistics of cloud phase using different lidars. Also, the importance of these errors

relative to other expected errors, especially those from multiple scattering, should be better discussed.

In its current form, I do not advice publication of the paper. With some major revisions, the paper might prove relevant for publication. Below I list my recommendations on how to improve the paper.

General comments:

1. The structure of the paper is not optimal which makes it difficult to follow. Much of this could be corrected by moving section 3 forward, as terms such as "analog detection" and "photon counting" are described here, but already used earlier in the paper. Also, multiple scattering is discussed on page 12 and 16, but I suggest to bring this forward to convince the reader that the analysis would still be relevant even if multiple scattering is considered.

2. Although the issue of multiple scattering is discussed, in my opinion it needs to be quantified more. How do the increases in depolariation due to multiple scattering relate to the biases discussed in the paper?

3. On page 3, line 2 it is stated about polarimetric lidar that "If not properly designed or considered, measurements can be misinterpreted casting doubt on critical measurements like cloud phase." Are there references backing up this statement? Are there indications that phase is misinterpreted in previous studies because of design issues? The references on page 2, line 20 (and other papers) very consistently conclude that thin liquid clouds or mixed phase clouds, often with ice precipitation, are prevalent in the Arctic, at least in the summer (e.g., Intrieri et al., 2002). Thus, more discussion on previous studies is needed to place the current work into context.

I suggest to add a short review on measurements made by other lidar systems. The main question is how other lidars acquire depolarization? Is either "analog detection" and "photon counting" the norm? Are the errors shown here also expected for other

lidars? Are statistics from previous lidars biased and are they inconsistent because different lidar systems were used? What are typical dynamic ranges for lidars? Such a discussion would improve the relevance of this paper. This context should also be repeated in the conclusions and abstract.

4. Much of the analysis in section 2 is in terms of "count rates". Could you please relate/convert this quantity to any physical value that lidar users might be accustomed to? It is also confusing how count rate relates to voltages measured in the Analog measurements. I cannot judge whether large depolarization errors at a given high count rate of, e.g., 100 MHz, are of any concern. Count rate probably relates to backscatter coefficient with units of per meter per steradian, so, if possible, convert it to that quantity. In addition, please discuss in which situations 'high' count rates are to be expected. It is stated later that in regions of multiple scattering, low count rates are expected because of attenuation. However, it seems that high count rate implies dense clouds and hence substantial multiple scattering.

5. Statistics of CAPABL data using different processing approaches are shown in 5, 6, 7 and discussed in section 5. Obvious differences are seen between the results with different measurements techniques. However, there is no objective verification of which of the results is most realistic. For example, it is stated that "there is a dramatic underestimate of liquid water by CAPABL's PC acquisition, which worsens with decreasing altitude, shown in Fig. 5." (page 12, last line), but this implies that one of the other acquisition approaches is "correct". There might be a good reason for assuming the analog method is best here, but please explain this reasoning before presenting these results. Figure 6 then presents results in terms of cloud height statistics. Again, the differences are clear, but I do not see how these are validations of the methods. Both liquid and ice can be present below ~4km, depending on temperatures (Intrieri et al, 2002), so all the statistics seem plausible to me. I suggest to remove this figure. Figure 7 suffers from the same problem. There are differences, but no indication of which result is most realistic. Please provide a more appropriate validation of the

measurements.

6. Section 2.2 offers a rather technical discussion on dynamic range. As an example, measurements in clear air are used. It is stated that" to measure clear air from 50m to 10 km would require no less than 6 orders of magnitude." These "clear air" measurements are of course very different form the targeted measurements in clouds. It is unclear how this discussion relates to measurement requirements and errors for cloud cases. Is this a best or worst case scenario? Elsewhere in the paper (page 17), optically thick clouds are referred to as "high dynamic range targets", so could we expect larger dynamic ranges?

Specific comments:

In equation 1, should the transmission not be a function of wavelength too?

Page 4, line 19: Please place equation 2 here in the text, embedded in a sentence. Please use this common style of equations embedded in sentences throughout the paper.

Page 4, line 20: Please define what is meant with "tilt angle". Also, I suggest to change "polarization" to "polarized signal" in this sentence (in 2 places).

Page 5, line 4: I am not sure what is meant with "which is not a function of range but only receive polarizations". Is there a part missing? Please correct.

Page 5, Eqs. 4, 5, 6 and 7. Please embed these in a sentence.

Page 6: I am puzzled what is meant with: "If one were to consider the differences in signal strength of a depolarization ratio of 1% vs 100%, there is substantial variations as well." Please rephrase.

Page 6 and other places: I suggest to add "of signal" after "orders of magnitude" everywhere in the paper.

Section 2.3: Please cite the paper by Gimmestad (Appl. Opt. 47, 3795-3802,2008) and

relate the definitions of depolarization used here to those discussed by Gimmestad.

Page 7, line 6: Here it is stated that "These are the theoretical arrival rates and not the observed arrival rates." This is a confusing statement, since the ones described in Eq. 9 are still theoretical. Please rephrase this statement.

Page 7: Please define "non-paralyzable system"

Page 7, line 24: Please note in the paper that the threshold of 0.11 is very arbitrary. Theoretical values for most ice crystals are much higher (e.g., Noel et al, Appl. Opt., 41, 4245– 4257, 2002). Also, mixing of aerosols could lead to misclassifying ice cloud (precipitation) as liquid, as shown by Bourdages et al. (Atmos. Chem. Phys., 9, 6881–6897, 2009) and Van Diedenhoven et al. (J. Appl. Meteorol. Climatol., 50, 2184-2192, 2011). Values around 0.1-0.2 might be commonly used, but they are not very robust.

Page 8: last sentence: What are "noise mitigating discriminators"? Is there a reference for this?

Page 9, line 10: I assume the range should be 0.03-0.11.

Page 10, last sentence: Please define backscattering ratio. Is backscattering ratio also affected by the measurement method?

Page 11, line 3: The backscatter cut-offs (26, and 50) are mentioned here, but discussed in the next section. Please move the discussion upward. I suggest to rename the "sub-visual" class to "aerosols" as it is confusing later when it is stated that these do not contain any ice or liquid. In any case, the distinction between clear-sky and subvisual is not very clear and might be very arbitrary. I would suggest combining them.
* * *

---

## Referee Comment (RC2) · Anonymous Referee #4 · 20 Mar 2017

**Review of "Low-Level, Liquid-Only and Mixed-Phase Cloud Identification by Polarimetric Lidar" by Stillwell et al.**

The manuscript discusses application of the CAPABL lidar at Summit, Greenland to retrieve cloud phase. Accurate identification of cloud phase is important for constraining the Arctic energy budget, but is a challenge for remote sensors. A polarization-sensitive lidar is able to discriminate ice from liquid cloud since ice particles are depolarizing, while liquid particles are not. However, it is well known that multiple scattering in optically thick liquid clouds can bias the measured depolarization ratio, making liquid clouds appear as if they were ice. Multiple scattering effects are not explored in this paper. Instead, this work focuses on two aspects of the CAPABL lidar (results succinctly summarized on Pg. 15, Ln. 22-28):

> 1) using an additional polarization channel at 45$^\circ$ together with the traditional parallel (0$^\circ$) and perpendicular (90$^\circ$) channels to improve signal strength both in the clear air below cloud and in the region above cloud where the signal is weak due to significant attenuation
>
> 2) comparison of photon counting and analog signals for extending the dynamic range of the detector and for assessing similarities and differences in the cloud phase discrimination map (Figures 3 and 4).

Summary statistics for a 4-month period show that the altitude profiles for the analog and photon counting signals are similar for both ice and clear air pixels, but large differences are observed for liquid pixels (Figures 5 and 6).

The manuscript is quite long and full of technical details that do not seem necessary to understand the results and discussion, and hence detract from readability. This makes is hard to assess the novelty of the work, since much of the data processing and analysis seems to follow directly from Neely et al., 2013. Depolarization ratio is still the major discriminator between ice and liquid, and the threshold value of $\delta$ = 0.11 is unchanged from Intrieri et al., 2002. This is not helped by a weak conclusions section – the "three key points" are vague and do not offer the reader much guidance as to how this work would inform their efforts to use polarization-sensitive lidar to discriminate ice from liquid (as discussed below). In sum, while a revised version of this paper may be publishable in AMT, the current technical scope (focused on detectors and photon count rates) seems more appropriate for a more specialized optics journal. The authors would also need to address the novelty of the revised submission in the context of the prior instrument paper (Neely et al., 2013).

***Specific Comments:***

1. Section 2 (Polarization Theory) is superfluous and should be removed. The few equations and definitions that could be considered relevant to the data analysis (e.g., Eqns. 6, 7, 8, and maybe 9) can easily be rolled into Section 4 (Data Analysis and Cloud Phase Identification).

2. Throughout the paper, the authors repeatedly bring up interesting science questions that are important for the data analysis and interpretation, but then summarily dismiss these considerations as beyond the scope of this paper. This sort of writing is weak, and the paper (and its scientific contribution) would be made all the much better if the authors were to delve more deeply into these issues. Since, in my opinion, the current scope of the paper is not

necessarily worthy of publication, tackling some of these issues in a novel way would improve my review of the paper. Specific topics include:

    a. **Constant bias in detector signal associated with multiple scattering, Pg. 12, Ln. 14-20:** "There are many techniques to deal with multiple scattering including multiple field of view lidar systems or post processing tools like those used by Shupe (2007), which reclassify shallow ice layers identified at the top of mixed-phase or liquid-only layers as mixed phase or liquid. For this analysis, multiple scattering clearly skews some of the interpretations towards ice but as the signals from A, PC, and SCPC are all subject to the exact same detector signals, the effect is consistent across all 3 data sets. This results in a constant bias for all three detection methods but as the purpose of this paper is to examine differences between the data sets, multiple scattering is recognized for future work but not implemented in the masking scheme."

    b. **Optimum combination of orthogonal/non-orthogonal depolarization channels, Pg. 14, Ln., 30-32:** "The results of this work highlight the differences in signal dynamic range that propagate through the provided analysis altering the physical interpretation of the measurements made. While combining the measurements into the optimum combination of signals is beyond the scope of this work, it is useful to broadly understand the way to combine all the different signal approaches to utilize the available data to extend the work started with CAPABL to different lidar systems"

    c. **Signal depolarization caused by multiple scattering of liquid droplets, Pg. 16, Ln. 23-32:** "One of the major topics to discuss is the handling of multiple scattering. Multiple scattering tends to increase signal strength but is important primarily within regions of high optical thickness. Even with scatterers that are purely spherical, multiple scattering can cause signal depolarization. In the CAPABL data set, this is most noticed in the middle and top of low-level liquid-only and mixed-phase clouds. The focus of this paper has been differences caused by count rate and signal strengths…The effect of multiple scattering is suggested for future work to further refine the measurement capabilities of CAPABL."

3. The discussion on "gluing" at the top of Pg. 15 seems unnecessary since this method is not actually applied in this paper.

4. The conclusions section summarizes the results of the manuscript in terms of "3 key points" that are demonstrated by this work:
    a. "cloud phase classification by polarimetric lidar is sensitive not only to the cloud phase but other cloud properties such as base height (or range) and optical depth, and to lidar design properties such as the power aperture product, field of view, receiver polarization and detection schemes."
    b. "this associated signal diversity in the lidar observations must be recognized in order to flag conditions unsuitable for determine cloud phase, an inherent problem in two-channel polarization lidars."

c. "by employing multiple planes of polarization in the lidar receiver, in the case of CAPABL four linear planes, the diversity in backscattered intensity may be handled more judiciously making the characterization of cloud types more accountable."

Regarding Pt. A, there is no discussion of how lidar design properties influence cloud phase classification in this manuscript. The conclusion is the first place that power aperture and field of view are mentioned in the manuscript. I don't follow how cloud base height influences the cloud phase classification – I would think that the signal attenuation and the range of the feature of interest would be much more important than if the cloud base is at, e.g., 500 m or 1000 m.

Regarding Pt. B, I'm not sure what this key point means, nor why two-channel polarization lidars are particularly problematic. Recognizing signal diversity in order to flag is not a particularly strong finding.

Regarding Pt. C, I don't know what this means either. What is meant by the phrases can be "handled more *judiciously*" or makes "the characterization of cloud types more *accountable*"? As in Pt. B, this is not an important finding.

5. On Pg. 17, Ln. 29-30, it is reported that the polarization configuration and signal combination allow the instrument to self-analyze limitations in a channels performance and correct some of the behavior. How is this self-analysis and correction done?

6. The recommendations for future analysis on Pg. 18, Ln. 10-13 sound great, and it's disappointing that none of these efforts were included in this paper. Are there other ancillary measurements of this kind at Summit that can be used to independently evaluate the lidar retrievals and assess the accuracy of the cloud phase discrimination? If so, I would strongly encourage the authors to incorporate such data into evaluating their lidar retrievals.

7. The terminology in Figures 3 and 4 is confusing and requires clarification on what exactly is being presented. I assume that "Total Backscatter" is really the "Total Attenuated Backscatter" or has an inversion been applied here beyond just adding the two channels to each other? Similarly, the label "Depolarization ($F_{33}$)" seems inconsistent with $d$ as in Eqn. 6, and the same inconsistency seems to apply for "Diattenuation ($F_{12}$)" and $D$ in Eqn. 7. It's unclear what is meant by Backscattering Ratio (e.g., ratio of backscatter coefficient to molecular scattering coefficient, or ratio of attenuated backscatter coefficient to molecular scattering coefficient) and how the inversion technique of Klett (1981) was applied here – does the inversion account for both particle and molecular attenuation or just the molecular? If particle attenuation is removed, then how was the inversion carried out (e.g., starting at high altitude or low altitude)? What lidar ratios were assumed? Last, it would be helpful to have the units for all of these graphs, and to report backscatter coefficient in terms of the more traditional $km^{-1}$ $sr^{-1}$ rather than photon count rate.

8. At the end of the day, what key finding or recommendation or technique is provided by this paper that allows someone like me to better employ a polarization-sensitive lidar to accurate determine cloud phase? How does the technique employed here compare to, or improve upon,

the cloud phase retrieval techniques employed by other polarization-sensitive lidars, e.g., the CALIOP lidar?

9. The author contributions statement on Pg. 19, Line 20 reads: "R. Stillwell prepared the manuscript with contributions from all co-authors." The brevity and lack of detail in this statement is completely unacceptable. Based on the acknowledgement of an NSF GRFP Fellowship, presumably the first author is a student so I would expect to see someone with the contribution of advising and supervising the research. Similarly, who took the data? Who maintained the instrument? Who analyzed the data? Why is this a 5-author paper?

**Minor Comments:**

1. In Figure 2, the y-axis is incorrectly labeled depolarization instead of depolarization ratio.
2. It's hard for me to interpret Figure 7 other than to note that PC seems to be seeing liquid clouds less often than the Analog, and SCPC is similar or in between. Which is correct?
3. Appendix A and Figure 8 are not meaningful. I suggest that this section be removed or moved to the Supplementary Material.

---

## Author Comment (AC1) · 1 May 2017

**Author's Response to Reviewers Comments (Review 1)**

**Title:** Low-Level, Liquid-Only and Mixed-Phase Cloud Identifications by Polarimetric Lidar

The authors would like to first thank the reviewers for their thoughtful and constructive comments on our manuscript. After discussion with coauthors, it was decided to complement this manuscript submission with a second manuscript to AMT. We realized after this review that the papers are not two manuscripts but two parts of the same. This current manuscript focuses on the observations and classification scheme while the second manuscript combines all the possible lidar observations into a best estimate cloud product and validates this product with co-located sensors resulting in a unique interpretation of cloud effects on the surface radiation budget. As such, we have changed the title of this manuscript and have included results from the second manuscript in this document to address the comments from this review of the first manuscript. The title of the companion manuscript is "Identifying and Characterizing the Properties of Arctic Clouds Using Enhanced Polarimetric Lidar. Part 2: Data Merging and Interpretation", and was submitted to AMT (on April 13, 2017) and should be accessible soon through the online portal.

The comments, taken from the provided reviews, have been copied in bullet format and addressed in the sub-bullets. A draft of the changes is also included where omissions are marked with  and additions are given in blue.

- Comment
  - Response

Reviewer 3

- The submitted manuscript does not provide an objective validation about which measurement strategy is more accurate
  - Owing to the uniqueness of the polarimetric lidar data set, the need for a careful evaluation of signals, and the unique cloud conditions over Summit Greenland, this first paper focuses on methods to best retrieve cloud properties in the context of the lidar's four polarization states using both photon and analog signal detection schemes. This measurement arrangement provides an overdetermined condition for estimating cloud phase, thus, enabling an internal validation of signals leading to optimal selection for science investigations. An internal validation is achieved by understanding the limitations of these signals and how these limitations may impact the estimate of cloud properties. This must be known before expanding the measurements to an objective validation of the technique and before using the results for scientific investigation. By taking this approach, this first manuscript elucidated common pitfalls of traditional lidar measurements in characterizing cloud properties, demonstrated a more optimal approach using non-orthogonal polarization measurements, generalized the theory of Neely et al. 2013, and provided a more definitive means of identifying what is a system effect and what is an effect associated with cloud properties. These findings enable the next step of performing an objective validation of the CAPABL data product. To address this need, the second manuscript, identified as "Part 2: Date Merging and Interpretation", has been submitted in parallel with this paper. The Part 2 manuscript invokes co-located remote sensing instrumentation to provide the objective validation and interpretation of the lidar results with new findings on cloud effects on surface radiation budgets at Summit.
- The submitted manuscript does not adequately discuss the implication that this work might have on previous statistics of cloud phase using different lidars. I suggest to add a short review on

measurements made by other lidar systems. The main question is how other lidars acquire depolarization? Is either "analog detection" and "photon counting" the norm? Are the errors shown here also expected for other lidars? Are statistics from previous lidars biased and are they inconsistent because different lidar systems were used? What are typical dynamic ranges for lidars? Such a discussion would improve the relevance of this paper. This context should also be repeated in the conclusions and abstract.

- o The manuscript demonstrates common issues/errors that arise in cloud property estimates using conventional lidar systems that employ two-channel, orthogonal polarizations measurements with either photon counting or analog signal detection. The demonstrated issues/errors are not testable using such conventional systems because there are insufficient measurement options to identify system effects versus geophysical effects. Consequently these systems cannot identify or quantify when their derived cloud properties are biased or in error. Our system setup provides the unique opportunity to test common assumptions and to optimize the system performance for improved confidence in cloud property retrievals. For example, the misidentification of liquid-only clouds as ice due to different system effects on the co-polarized and cross-polarized is a problem that our system can identify and mitigate.

  Due to varying configurations and approaches by other lidars, we cannot specifically identify how well other systems have represented cloud properties. However, we identify potential shortcomings and introduce methods for other lidar systems to employ to identify and mitigate these system effects. Our goal is to provide a careful and complete analysis in this Part 1, by which, the second manuscript, Part 2, can leverage to study the derived cloud properties on the surface radiation budget.

- The importance of the measurement errors relative to other expected errors, especially those from multiple scattering, should be better discussed.
  - o A slight bias due to multiple scattering is observed in Figure 6, but as mentioned in the text in Section 4.3 it is common to all observation types made by CAPABL. Multiple scattering does not affect the implications of these measurements because the measured values all respond similarly. However, multiple scattering does cause changes in the best estimate cloud product and is directly addressed in the second manuscript, Part 2. By directly comparing CAPABL measurements to a co-located micropulse lidar with a $100 \ \mu rad$ field of view over a 6 month trial period where both systems have better than 94% uptime, the bias described can be measured. That paper concludes that fewer than 5.5% of cloud liquid voxels are affected by multiple scattering.

- Suggest bringing the definitions of analog and photon counting and multiple scattering forward from Section 3.
  - o The authors agree with the reviewer. Multiple scattering has now been inserted in Section 2.3 and is still addressed in 4.3. The definitions of analog and photon counting have now been inserted in Section 2.2 and are still further addressed in Section 3.

- Although the issue of multiple scattering is discussed, in my opinion it needs to be quantified more. How do the increases in depolarization due to multiple scattering relate to the biases discussed in the paper?
  - o Multiple scattering serves to bias all channels and thus cannot be uniquely isolated and removed without more information. Multiple scattering is addressed therefore in part 2 of this manuscript where CAPABL is directly compared to a lidar with much smaller field of view where the multiple scattering is suppressed. Over a 6-month period used for validation, multiple scattering affects fewer than 5.5% of all cloud voxels and does not affect the column data product at all (upon which Figure 7 is based). Multiple scattering tends to increase depolarization and causes liquid cloud tops to look like ice. As such, the column of data still contains liquid in every case and is thus tagged as liquid for the

column data product used in Figure 7 for example. The voxel height results presented in Figure 6 contain a small bias, less than 5% but this bias is constant across all 6 measurement types and as such does not affect the conclusion about altering medians due to saturation.

- It is stated about polarimetric lidar that "If not properly designed or considered, measurements can be misinterpreted casting doubt on critical measurements like cloud phase." Are there references backing up this statement? More discussion on previous studies is needed to place the current work into context.
  - The authors have added citations to Hayman and Thayer 2009 who find systematic depolarization bias due to optical retardance, Liu et al. 2009 who claim improvements in wind and aerosol measurements by considering saturation, and Neely et al. 2013 who consider saturation in their calculation of diattenuation.
- Much of the analysis in section 2 is in terms of "count rates". Could you please relate/convert this quantity to any physical value that lidar users might be accustomed to?
  - Count rates are defined in our appendix equation A2 relative to the raw lidar signals. Furthermore, saturation behavior is defined in terms of count rate by other authors (ex: Donovan 1993). The authors define signals in terms of count rates to facilitate a simple calculation of per shot photons per second making it a value independent of our specific system. This allows direct comparison between systems without having to consider different system specifications. The authors know of no other way to generalize signal levels in terms of observational values and related to saturation.
- Please discuss in which situations 'high' count rates are to be expected.
  - Many scenarios exist but in the context of this work, high count rates and correspondingly high dynamic range, occur primarily in low liquid cloud environments due to their small range and efficient scattering. A paragraph at the end of Section 4.1 is added to describe where one would expect saturation issues related to high count rates to appear.
- Obvious differences are seen between the results with different measurements techniques. However, there is no objective verification of which of the results is most realistic.
  - Our overdetermined measurement scheme allows for internal verification of which signals are being biased and which are not. For example, a different combination of signals is determined to measure high thin clouds, low thick clouds, or clear air, which all produce very different signal intensities especially when considering different polarizations. A best estimate data product is developed and validated in Part 2 of this work but is defined based on the results of the internal validation presented in this paper. The Part 2 paper then provides the objective verification using multisensor observations to confirm the lidar cloud classification scheme.
- Figure 6 then presents results in terms of cloud height statistics. Again, the differences are clear, but I do not see how these are validations of the methods. Both liquid and ice can be present below 4km, depending on temperatures (Intrieri et al, 2002), so all the statistics seem plausible to me. I suggest to remove this figure. Figure 7 suffers from the same problem as Figure 6.
  - The authors include Figure 6 and 7 to quantitatively assess the differences the 6 introduced processing methods can have on signals scattered from the same volume. This internal validation effectively establishes a direct link between lidar processing methods and errors in attribution of cloud properties presented in Section 6.2. The quantification of these differences is required to define the best possible method to optimize from the overdetermined six signal approach. The results of Figure 7 illustrate where certain signals are deficient and how a best estimate can be retrieved. For example, analog detection systematically underestimates cloud fraction (too much clear air compared to the best estimate product defined in Paper 2) and photon counting systematically

underestimates liquid cloud types (compared to the best estimate product defined in Paper 2)). Without the available signal options, none of the types of data presented are without bias given the variety of cloud types experienced at Summit.

An extension of Figure 7 from the text is shown here using the best estimate product from paper 2 to support the above statements.

[Figure]

- Section 2.2 offers a rather technical discussion on dynamic range. As an example, measurements in clear air are used. It is stated that "to measure clear air from 50m to 10 km would require no less than 6 orders of magnitude." These "clear air" measurements are of course very different form the targeted measurements in clouds. It is unclear how this discussion relates to measurement requirements and errors for cloud cases. Is this a best or worst case scenario? Elsewhere in the paper (page 17), optically thick clouds are referred to as "high dynamic range targets", so could we expect larger dynamic ranges?
  - Clear air is used and described because it provides a smooth transition from high signal strength to low signal strength in a known and quantifiable way-meaning the scattering and its behavior with altitude is constrained. Clouds, on the other hand, can produce the same range of signal strengths as clear air but can vary from voxel to voxel due to unknown scattering properties – in fact, this is the very thing we are trying to quantify. By using clear air we can understand these signal ranges and system responses while having a firm understanding of the scattering. For example, air molecules at these wavelengths produce very little depolarization, less than 1%. Thus, a measure of depolarization from clear air gives us a clear determination of the system's depolarization performance. Depolarization of 1% , for example, indicates some weak system depolarization. For CAPABL it is expected that the system induced depolarization is approximately 0.8%. However, this number is tolerable given the thresholds we impose on depolarization for estimating cloud type.  For example, liquid cloud targets have

depolarization values of approximately < 11% in the CAPABL data set. A clear 1-2 orders of magnitude of polarization-induced dynamic range is involved. If the depolarization is to be estimated for such low values, then the co-polarized and cross-polarized signals must differ by 2 orders of magnitude. This imposes a constraint on which signals are best to accommodate this range in signals. The signal dynamic range is compounded further by the variety in cloud base heights leading to range-dependent signal strengths along with varying cloud optical depths. The clarifying sentence: "Note that clear air and liquid bearing clouds both have high polarization-induced dynamic range, i.e. low depolarization ratio, but liquid bearing clouds have more rapid attenuation of signal, which is not strictly range-induced dynamic range but rather attenuation from transmission terms of the SVLE" is now included in Section 2.2

- In equation 1, should the transmission not be a function of wavelength too?
    - The reviewer is correct that transmission is a sensitive function of wavelength. This is indicated in the equation by the dependence on the wave vector that points in the direction of light propagation and has a magnitude equal to the wavenumber $\left|\bar{k}\right| = 2\pi/\lambda$.
- Page 4, line 19: Please place equation 2 here in the text, embedded in a sentence. Please use this common style of equations embedded in sentences throughout the paper.
    - The equation is now imbedded. Equations 2, 3, 4, 5, 6, 7, 8, and 9 and Appendix A1, A2, A3, and A4 have also been changed.
- Page 4, line 20: Please define what is meant with "tilt angle". Also, I suggest to change "polarization" to "polarized signal" in this sentence (in 2 places).
    - The authors agree that the statement is unclear. The reference to system tilt is removed as it is unnecessary. Details on the system orientation are given in Section 3. "Polarization" has been changed to "polarized signal".
- Page 5, line 4: I am not sure what is meant with "which is not a function of range but only receive polarizations". Is there a part missing? Please correct.
    - The sentence has been rewritten to clarify that that matrix A and inverse $A^{-1}$ are not functions of range. The A matrix is only a function of the receiver configuration.
- Page 5, Eqs. 4, 5, 6 and 7. Please embed these in a sentence.
    - These equations are now imbedded.
- Page 6: I am puzzled what is meant with: "If one were to consider the differences in signal strength of a depolarization ratio of 1% vs. 100%, there is substantial variations as well." Please rephrase.
    - This sentence has been removed.
- Page 6 and other places: I suggest to add "of signal" after "orders of magnitude" everywhere in the paper.
    - The orders of magnitude are now tied to those that are linked directly to the signal and those that are linked to the observing capacity of the system.
- Section 2.3: Please cite the paper by Gimmestad (Appl. Opt. 47, 3795-3802,2008) and relate the definitions of depolarization used here to those discussed by Gimmestad.
    - References to Gimmestad and references that are cited therein and references that cite and advance the definition of depolarization by Gimmestad are included in a paragraph at the end of section 2.1. Depolarization d used by the authors is consistent with the definitions presented by Gimmestad 2008, Flynn et al. 2007, and Hayman and Thayer 2009, 2012.
- Page 7, line 6: Here it is stated that "These are the theoretical arrival rates and not the observed arrival rates." This is a confusing statement, since the ones described in Eq. 9 are still theoretical. Please rephrase this statement.

<li>

<li>The sentence specified is modified to clarify that $S_{0_\perp}$ and $S_{0_{||}}$ are theoretical arrival rates where Equation 9 is used to relate theoretical arrival rates to possible observed depolarization errors based on theoretical models of saturation behavior.</li>

</li>
<li>Page 7: Please define "non-paralyzable system"

<li>A non-paralyzable counting system is specified as a theoretical model to link theoretical photon arrival rates to observed photon arrival rates. Additionally, a further note is now: "Note that the non-paralyzable model assumes that it takes some finite time for the photon counting system to reset before it can count another photon and is correct for CAPABL's photon counting system based on the analysis provided in Appendix A."</li>

</li>
<li>Page 7, line 24: Please note in the paper that the threshold of 0.11 is very arbitrary. Theoretical values for most ice crystals are much higher (e.g., Noel et al, Appl. Opt., 41, 4245– 4257, 2002). Also, mixing of aerosols could lead to misclassifying ice cloud (precipitation) as liquid, as shown by Bourdages et al. (Atmos. Chem. Phys., 9, 6881– 6897, 2009) and Van Diedenhoven et al. (J. Appl. Meteorol. Climatol., 50, 2184-2192, 2011). Values around 0.1-0.2 might be commonly used, but they are not very robust.

<li>The authors have re-analyzed the entire data set presented with depolarization thresholds from 0.05 to 0.30 with 0.01 spacing. Plotted below is the fractional occurrence of liquid and ice measured for July 2015 from the analog detection channel. Above approximately $\delta_O = 0.11$, the fractional occurrence stabilizes until approximately $\delta_O = 0.20$. Beyond that point, ice clouds are being lumped into the water fractional occurrence. Any value $0.11 \leq \delta_O \leq 0.20$ will yield similar conclusions for fractional occurrence change. From this we conclude that $\delta_O = 0.11$ is a reasonable threshold to use for the CAPABL data set and based on available literature.</li>

</li>

The Bourdages et al. reference is compelling and certainly worth considering. Over the 4-month data period presented, it doesn't appear that any precipitating hydrometeors (defined using the mean Doppler velocity as measured by a co-located millimeter cloud radar) are classified as liquid. Additionally, the findings of papers such as Morrison et al., Nature Geo., 5, 1, 11-17, 2012 suggest that in the Arctic liquid hydrometeors are too small to be efficiently precipitated. The validation study in Part 2 of this manuscript confirms that the radar measured mean Doppler velocity of liquid is lower than both randomly and preferentially oriented ice by a factor of at least 1.5.

[Figure]

- Page 8: last sentence: What are "noise mitigating discriminators"? Is there a reference for this?
  - A reference to Donovan et al. 1993 is included, which describes the added SNR advantage of photon counting.
- Page 9, line 10: I assume the range should be 0.03-0.11.
  - The authors agree. We have changed this typo.
- Page 10, last sentence: Please define backscattering ratio. Is backscattering ratio also affected by the measurement method?
  - Backscattering ratio is now defined in the text as the ratio of total to molecular scattering. The backscattering ratio is dependent on the derivative of the lidar signal, which is susceptible to saturation. Strong saturation causes bleaching (here we mean severe under-representation or low bias) of backscatter ratio due to a nullification of the derivative of interest needed for the Klett type inversion. This is observed frequently by photon counting methods and is discussed in detail in the validation study presented in Part 2 of this work especially related to low level (lower than 200-300 meters) fog and clouds.
- Page 11, line 3: The backscatter cut-offs (26, and 50) are mentioned here, but discussed in the next section. Please move the discussion upward.
  - Following the reviewer's suggestion, all the bounds and the rational for the bounds are now combined and listed in Section 4.2. Section 4.1 has been renamed to "Processing Methods".
- I suggest to rename the "sub-visual" class to "aerosols" as it is confusing later when it is stated that these do not contain any ice or liquid. In any case, the distinction between clear-sky and subvisual is not very clear and might be very arbitrary. I would suggest combining them.
  - Clear air and the sub-visible mask have now been combined. Figures 3, 4, and 7 now reflect this change.

- Much of the data processing and analysis seems to follow directly from Neely et al., 2013. The authors would also need to address the novelty of the revised submission in the context of the prior instrument paper (Neely et al., 2013). Section 2 (Polarization Theory) is superfluous and should be removed. The few equations and definitions that could be considered relevant to the data analysis (e.g., Eqns. 6, 7, 8, and maybe 9) can easily be rolled into Section 4 (Data Analysis and Cloud Phase Identification).
    - The expressions given in Section 2 are more general, and therefore more broadly applicable, than the analysis presented by Neely et al. 2013. Both are based on the theory of Hayman and Thayer 2012. The processing presented by Neely et al. 2013 cannot accommodate non-orthogonal retrievals as applied here but can be retrieved easily from this generalization. This is clarified with the sentence: "The expressions given in Eq. 6 and Eq. 7 are generalizations of the equations presented by Neely et al. (2013) that assumed fixed receiver polarization angles." Given that the analysis does not follow trivially from Neely et al. 2013, the authors believe it should be clearly stated.
- Throughout the paper, the authors repeatedly bring up interesting science questions that are important for the data analysis and interpretation, but then summarily dismiss these considerations as beyond the scope of this paper. This sort of writing is weak, and the paper (and its scientific contribution) would be made all the much better if the authors were to delve more deeply into these issues.
    - Much of the requested science results are presented in Part 2 of this work. However, the combination of signals and best estimate data product to be used for science investigation are predicated on the close examination of the various signals presented in this part of the work. Our unique measurement scheme enables an improved cloud data product by over determining the estimate of depolarization and diattenuation. The best estimate data product cannot be defensibly defined without carrying out the detailed optimization presented in this work, and have not been specified in the literature.
    - We have included in Section 6.2 a discussion of how improper cloud classification can cause erroneous estimates of radiative forcing. For example, using photon counting observations improperly can cause an error in longwave cloud radiative forcing of approximately $10W/m^2$. Miller et al. (2015) finds an average of $33W/m^2$ for cloud radiative forcing at Summit suggesting that using uncorrected CAPABL data to infer radiative impacts could under-represent forcing by as much as one third.
- Since, in my opinion, the current scope of the paper is not necessarily worthy of publication, tackling some of these issues in a novel way would improve my review of the paper. Specific topics include: a) Constant bias in detector signal associated with multiple scattering, Pg. 12, Ln. 14-20: b) Optimum combination of orthogonal/non-orthogonal depolarization channels, Pg. 14, Ln., 30-32: c) Signal depolarization caused by multiple scattering of liquid droplets, Pg. 16, Ln. 23-32.
    - The effect of multiple scattering and optimum combination of receiver signals are both addressed in Part 2 of this manuscript. It is the authors' opinion that multiple scattering cannot be defensibly analyzed without removing the confounding effects related to saturation, which are calculated in this work. Similarly, the optimum combination of signals is based on the signal count rate limitations shown here to cause biases in cloud voxel height and fractional occurrence. The findings of Part 2 are: multiple scattering causes biases on the order of 5% of cloud voxels whereas saturation induced errors are upwards of 30%, and that the merged data product results in 27% increase in observable voxels using all 6 types of signals defined here with no net phase bias as compared to a co-located micro-pulse lidar.

- The discussion on "gluing" at the top of Pg. 15 seems unnecessary since this method is not actually applied in this paper.
  - The author's agree. The section has been removed.
- There is no discussion of how lidar design properties influence cloud phase classification in this manuscript. The conclusion is the first place that power aperture and field of view are mentioned in the manuscript. I don't follow how cloud base height influences the cloud phase classification – I would think that the signal attenuation and the range of the feature of interest would be much more important than if the cloud base is at, e.g., 500 m or 1000 m.
  - The Stokes vector lidar equation presented in Section 2 is the major linking feature to design parameters. We show that base height is linked to depolarization errors through its impact on receiver dynamic range. As the cloud base varies, the saturation in photon counting detection varies and is not constant for a given count rate or depolarization ratio as shown in our figure 2. The sentence referenced by the reviewer has been modified to clarify that count rate links directly to depolarization ratio errors, which have non-negligible effect on cloud classification.
- I'm not sure what this key point means, nor why two-channel polarization lidars are particularly problematic. Recognizing signal diversity in order to flag is not a particularly strong finding.
  - The two channel polarization measurements are not particularly problematic but their interpretation of cloud properties can be unknowingly incorrect depending on the cloud conditions. The purpose of the paper is not to provide a quality flag but to illustrate and quantify under what conditions certain signals bias the estimate and how signal diversity can be employed to improve the cloud classification. Using multiple polarization measurements overdetermines the estimate of depolarization and can then be used dynamically and autonomously to select the proper combination for the given cloud conditions. This increases the confidence in the retrieved cloud classification and improves the overall data availability by almost 27% (as discussed in Part 2 of this work). That is simply a result of the minimization of available dynamic range by selecting polarization components with very different signal dynamic ranges. Additionally, as diattenuation is a sensitive measure of saturation, it can be used to further improve cloud classification.
- What is meant by the phrases can be "handled more judiciously" or makes "the characterization of cloud types more accountable"?
  - Here we simply mean that depolarization can be of geophysical origin or a result of limitations of the observational system. We recognize that depolarization is a mix of both and we use the diattenuation product $D_1 D_2$ to identify where saturation is causing errors with depolarization measurements. Section 6.3 is added to clarify this point.
- On Pg. 17, Ln. 29-30, it is reported that the polarization configuration and signal combination allow the instrument to self-analyze limitations in a channels performance and correct some of the behavior. How is this self-analysis and correction done?
  - Self analysis relates to the systematic tagging of saturation via diattenuation. It is shown to be a sensitive indicator of saturation. The inversions in Section 2 show a need for only 3 polarization measurements. However, 4 measurements are made. The $F_{12}$ term of the scattering matrix is measured twice with retrievals of opposite sensitivity to saturation then multiplied together. Only non-zero diattenuation with the same sign is physical where the areas with negative product are caused by saturation. Voxels with negative diattenuation products are removed. In the case of saturation issues, the merging procedure presented in Part 2 uses the weakest signals using non-orthogonal retrievals to attempt to correct for saturation of the strongest channel. This correction stage is only necessary if the standard retrievals fail. This is clarified at the end of Section 4.1 and with the addition of Section 6.3.

- The recommendations for future analysis on Pg. 18, Ln. 10-13 sound great, and it's disappointing that none of these efforts were included in this paper. Are there other ancillary measurements of this kind at Summit that can be used to independently evaluate the lidar retrievals and assess the accuracy of the cloud phase discrimination? If so, I would strongly encourage the authors to incorporate such data into evaluating their lidar retrievals.
    - The recommendations presented in this paper are addressed in detail in Part 2 of this work. Co-located millimeter cloud radar, micropulse lidar, microwave radiometer, and radiation measurements are used to analyze the best estimate retrieval of CAPABL. By making the paper a 2 part series, these results are linked.
- The terminology in Figures 3 and 4 is confusing and requires clarification on what exactly is being presented. I assume that "Total Backscatter" is really the "Total Attenuated Backscatter" or has an inversion been applied here beyond just adding the two channels to each other? Similarly, the label "Depolarization (F33)" seems inconsistent with d as in Eqn. 6, and the same inconsistency seems to apply for "Diattenuation (F12)" and D in Eqn. 7. It's unclear what is meant by Backscattering Ratio (e.g., ratio of backscatter coefficient to molecular scattering coefficient, or ratio of attenuated backscatter coefficient to molecular scattering coefficient) and how the inversion technique of Klett (1981) was applied here – does the inversion account for both particle and molecular attenuation or just the molecular? If particle attenuation is removed, then how was the inversion carried out (e.g., starting at high altitude or low altitude)? What lidar ratios were assumed? Last, it would be helpful to have the units for all of these graphs, and to report backscatter coefficient in terms of the more traditional km-1 sr-1 rather than photon count rate.
    - The authors agree and have clarified the figures. Total backscatter is now listed as Relative Backscatter. No inversion is applied to correct for attenuation or range correcting.  Depolarization is now listed as d to stay consistent with the definition of Gimmestad 2008. Diattenuation is now consistent with the text where it is calculated twice and multiplied. Backscatter ratio is as calculated by Neely et al. 2013 and is labeled R. The calculation is listed in the text as well.
- At the end of the day, what key finding or recommendation or technique is provided by this paper that allows someone like me to better employ a polarization-sensitive lidar to accurate determine cloud phase? How does the technique employed here compare to, or improve upon, the cloud phase retrieval techniques employed by other polarization-sensitive lidars, e.g., the CALIOP lidar?
    - Ground based lidar systems observing the low troposphere like CAPABL suffer from an enormous dynamic range resulting from the solid angle term of the Stokes vector lidar equation that is not experienced by space based lidar systems. As highlighted in section 2.2, the range from 50 meters to 5 km is 2 orders of magnitude greater than from 10 km to 100 km. All else being equal, this places a higher demand of the observing system because range-induced dynamic range is a more substantial fraction of the overall dynamic range. This work highlights that non-orthogonal polarization retrievals can reduce the dynamic range requirement of polarization lidar signals, which enhance their overall range. Our findings illustrate how adding at least a 3[rd] and more ideally a 4[th] polarization channel can be used to understand when saturation is affecting measurements and quantifies the effect. Only by adding more polarization channels can systematic effects be measured that confound the measurements of backscattering and depolarization made by 2 polarization systems. Paragraph 2 of Section 7 has been modified to clarify these points.
- The author contributions statement on Pg. 19, Line 20 reads: "R. Stillwell prepared the manuscript with contributions from all co-authors." The brevity and lack of detail in this statement is completely unacceptable. Based on the acknowledgement of an NSF GRFP

Fellowship, presumably the first author is a student so I would expect to see someone with the contribution of advising and supervising the research. Similarly, who took the data? Who maintained the instrument? Who analyzed the data? Why is this a 5-author paper?

- o The authors agree with the reviewer. The authors have included a more extensive list of author contributions.
- In Figure 2, the y-axis is incorrectly labeled depolarization instead of depolarization ratio.
  - o The figure y-axis label has been updated as suggested.
- It's hard for me to interpret Figure 7 other than to note that PC seems to be seeing liquid clouds less often than the Analog, and SCPC is similar or in between. Which is correct?
  - o There are different effects that can impact each of the 6 measurements differently. Figure 7 illustrates this by presenting all 6 measurements used to classify the cloud type or clear air from the same scattering volume. A merged best estimate data product is determined by optimizing all of the available information from these six measurements. As described in earlier sections for example, analog detection systematically underestimates cloud fraction (too much clear air) and photon counting systematically underestimates liquid cloud types. With proper evaluation and multiple polarization channels, each cloud type and observing condition can be evaluated, biases quantified, and cloud products optimized for application to scientific investigation – as presented in Part 2.
- Appendix A and Figure 8 are not meaningful. I suggest that this section be removed or moved to the Supplementary Material.
  - o Appendix A is simply used to support the assertion that CAPABL is a non-paralyzable system with its deadtime and deadtime error bound and as a collection of relevant information for saturation analysis of lidar systems. The authors feel it is a needed repository of information. The authors want the presented data as transparent as possible so that others looking to implement such a scheme can be complete and, as such, have a preference for leaving the information as an appendix, but would be OK with its inclusion as a supplement if insisted.

**Identifying and Characterizing the Properties of Arctic Clouds Using Enhanced Polarimetric Lidar. Part 1: Observations and Classification**

Robert A. Stillwell[1], Ryan R. Neely III[2,3], Jeffrey P. Thayer[1], Matthew D. Shupe[4,5], and Michael O'Neill[4]

[1]Aerospace Engineering Sciences, University of Colorado at Boulder, ECNT 320, 431 UCB, University of Colorado, Boulder, CO 80309.
[2]School of Earth and Environment, University of Leeds, LS2 9JT, Leeds, UK.
[3]National Centre for Atmospheric Science, University of Leeds, LS2 9JT, Leeds, UK.
[4]Cooperative Institute for Research in Environmental Sciences, University of Colorado at Boulder, 216 UCB, University of Colorado, Boulder, CO 80309.
[5]Earth System Research Laboratory, National Oceanic and Atmospheric Administration, 325 Broadway, Boulder, CO 80309.

*Correspondence to:* Robert Stillwell (robert.stillwell@colorado.edu)

**Abstract.** The measurement of low-level, liquid-only and mixed-phase clouds in the polar regions is a necessary building block to understand the regional surface energy and mass budgets over ice sheets. The unambiguous retrieval of cloud phase from polarimetric lidar observations is dependent on the assumption that only cloud scattering processes alter the transmitted polarization. However, due to clouds varying in range, optical depth, and scatterer size and shape, most atmospheric lidar systems  experience high signal dynamic ranges, which can impact the lidar system response and make the cloud phase determination ambiguous. Due to the high optical thickness and predominately low-lying nature of liquid-only and mixed-phase clouds in the polar regions, relative to ice-only clouds, a systematic  bias of the traditional lidar depolarization ratio, which uses co-polarized and cross-polarized signals, can occur that is not always identifiable in traditional polarimetric lidar systems. For both liquid-only and mixed-phase clouds, this results in a misidentification of liquid water in clouds as ice, which has broad implications on evaluating surface energy budgets. The Clouds Aerosol Polarization and Backscatter Lidar (CAPABL) at Summit, Greenland employs multiple planes of linear polarization, and photon counting and analog detection schemes, to self evaluate, correct, and optimize signal combinations to improve cloud classification. Using novel measurements of diattenuation that are sensitive to both preferentially oriented ice crystals and counting system non-linear effects, unambiguous depolarization ratio measurements are possible by over constraining polarization measurements. This overdetermined capability for cloud phase determination allows for system errors to be identified and quantified in terms of their impact on cloud properties. Furthermore, the multiple signals permits optimal selection of signals to provide the best estimate of the cloud property. For example,  the difference in the estimate of the median height for liquid clouds is shown to differ by as much as  2

those identified as liquid due to a systematic bias in photon counting signals km between analog and photon counting detection because of photon-counting detection misidentifying the presence of low-lying liquid clouds as ice. At a constant altitude, more than 94% of the liquid pixels voxels identified with analog signals can be misidentified as ice with when using photon counting signals. This results in a possible error of fractional occurrence of cloud liquid the assessed radiative impact of liquid clouds of approximately 30% . It is shown when using traditional polarimetric lidar. Observations from CAPABL show that by observing polarization planes that are non-orthogonal , the dynamic range and adding measurements of diattenuation, saturation effects of observed signals is reduced, the coverage of the expected signal dynamic range is increased, and more linear response can be captured enabling unambiguous measurements of the lidar depolarization ratio from atmospheric scatterers. Using non-orthogonal polarization observations is shown to enhance measurement sensitivity, increasing the effective sampling range for CAPABL by as much as 18% or approximately 1.5 km in clear air.

**1 Introduction**

[revised manuscript text omitted]

15  stringent requirement of linear signal operation over a large dynamic range. If not properly designed or considered, measurements can be misinterpreted casting doubt on critical measurements like cloud phase  (Hayman and Thayer, 2009; Liu et al., 2009; Neely et al., 2013). For example, traditional two-channel orthogonal polarization measurements using co-polarized and cross-polarized signals can not unambiguously separate systematic polarization effects and geophysical effects. These measurement errors  result in cloud misidentification, which, in turn, introduce unquantified error into model results which

20  are used to study key cloud and radiative processes. Observations by lidar of polar liquid-only and mixed-phase clouds in particular are challenging due to their high optical thicknesses, relative to ice-only clouds, and low-lying altitude, which demands large system dynamic ranges.

This work focuses on novel polarimetric lidar measurements made at Summit, Greenland  (72°35′46.4"N, 38°25′19.1"W, 3.212 km amsl) as part of the Integrated Characterization of Energy Clouds Atmospheric

25  State and Precipitation at Summit (ICECAPS) program outlined by Shupe et al. (2013). The measurements to be presented are taken from the Clouds Aerosol Polarization and Backscatter Lidar (CAPABL), which was originally designed to measure polarization properties of clouds with emphasis on identifying preferentially oriented ice crystals and cloud phase (Neely et al., 2013). Analysis of five years of data observed by CAPABL has highlighted several uncertainties and biases that can cause errors in the interpretation of geophysical retrievals of cloud phase, primarily caused by the system limitations to adequately

30  handle the observable dynamic range in backscattered signals from clouds.  This paper serves as Part 1 in a two-part series to definitively describe these errors  by employing CAPABL's advanced polarization capabilities. Furthermore, this Part 1 paper demonstrates how CAPABL can be used to improve cloud property retrievals important for scientific study. The Part 2 paper exploits these advancements to optimize cloud property estimates and, with complementary multi-sensor

35  observations at Summit, determine the effects of the cloud properties on the surface radiation budget.

The outline of this paper is as follows. The  measurement theory used, upon which the retrievals within CAPABL's automatic processing are based, is stated in Sect. 2. An overview of the CAPABL system is provided in Sect. 3 with an emphasis on the significant updates that have been made to the instrument since its introduction by Neely et al. (2013). An overview of the data processing is provided in Sect. 4 with emphasis on geophysical retrievals and potential errors caused by  signal dynamic range. Relevant cloud statistics for a 4 month period are compiled in Sect. 5. A comparison of results focusing on the errors of geophysical cloud property estimates resulting from data misinterpretation is given in Sect. 6. Finally this paper concludes in Sect. 7 with a summary and recommendations for general cloud observations via lidar and suggestions for additional work to further improve polarimetric lidar retrievals of polar liquid-only and mixed-phase clouds.

**2 Measurement Theory**

**2.1 CAPABL's Polarization Retrievals and Mueller Formalism**

Polarimetric lidar leverages the vector nature of light to more completely describe scattering. Using a vector description of light allows one to describe scatterers by how they alter polarization states of light as well as how much energy is redirected. Hayman and Thayer (2012) use polar decomposition of Mueller matrices to define the Stokes vector lidar equation (SVLE), which links transmitted and received polarization states of light to physical attributes of the scatterers. This equation forms the basis of CAPABL's polarization retrievals and is given in Eq. 1

$$\bar{N}(R) = \bar{\bar{O}}\bar{\bar{M}}_{R_x}\left(\bar{k}_s\right)\left[\left(G(R)\frac{A}{R^2}\Delta R\right)\bar{\bar{T}}_{atm}\left(\bar{k}_s,R\right)\bar{\bar{F}}\left(\bar{k}_i,\bar{k}_s,R\right)\bar{\bar{T}}_{atm}\left(\bar{k}_i,R\right)\bar{\bar{M}}_{T_x}\left(\bar{k}_i\right)\bar{S}_{T_x} + \bar{S}_B(\lambda_{R_x})\right] \tag{1}$$

where $\bar{N}$ is vector of photon counts for each polarization channel as a function of range, $R$, $\bar{\bar{O}}$ is the observation matrix describing each polarization observation channel, $\bar{\bar{M}}_{T_x}$ and $\bar{\bar{M}}_{R_x}$ are the Mueller matrices describing the transmitter and receiver, which are functions of the incident and scattered wave vector $\bar{k}_i$ and $\bar{k}_s$, respectively, $G$ is the physical overlap function of the transmitter and receiver, $A$ is the telescope area, $\Delta R$ is the range resolution of the counting system, $\bar{\bar{T}}_{atm}$ is the one way transmission Mueller matrix either between the transmitter and the scatterer or between the scatterer and the receiver, $\bar{\bar{F}}$ is the scattering phase matrix, which is a function of both transmitted and received wave vectors and range, $\bar{S}_{T_x}$ is the Stokes vector of the light from the laser source, and $\bar{S}_B$ is the Stokes vector of the background condition which is a function of the receiver wavelength window, $\lambda_{R_x}$. The terms of the equation are organized by their functional order because matrix operations do not generally commute. The observation matrix is also included because only intensity can be measured directly with the full Stokes vector determined through measurement with particular configurations of the analyzer (Hayman and Thayer, 2012). Finally, the standard assumptions used to derive the lidar equation including independent and single scattering are also used here, though they are not strictly required. For more information on the SVLE and its derivation, the reader is referred to Hayman and Thayer (2012).

Elements of $\bar{\bar{F}}$ can be used to describe physical attributes of scatterers beyond simple scattering cross section (Kaul et al., 2004). The reader is referred to Neely et al. (2013) who describe the polarization retrievals and the physical interpretation

of the elements CAPABL measures in detail.  The retrieval presented by Neely et al. (2013) is generalized here by relaxing the assumptions made in that work, namely that the receiver orientations are fixed at $0°$, $45°$, and $90°$ relative to the output linear polarization.

From this general form  given in Eq. 1, the number of photons to be observed can be derived in each polarization channel, given in Eq. 2  as

$$N_M(R) = \xi(R)\left[F_{11}(R) + \cos(2\theta)F_{12}(R) + \cos(2\phi)(F_{12}(R) + \cos(2\theta)F_{22}(R)) + \sin(2\theta)\sin(2\phi)F_{33}(R)\right] \tag{2}$$

assuming that CAPABL: 1) emits a linear  polarized signal at angle $\phi$  (yielding the simplification $\bar{\bar{M}}_{T_x}(\bar{k}_i)\bar{S}_{T_x} = \begin{bmatrix} 1 & \cos(2\phi) & \sin(2\phi) & 0 \end{bmatrix}^T$ ), and 2) only measures linear  polarized signal at angle $\theta$ from the reference transmit polarization, (Neely et al. (2013) Eq. 15 with $A(\Gamma_{wp}) = \bar{\bar{M}}_{R_x}(2\theta)$). These assumptions have been questioned for some optical systems, e.g. (Di et al., 2016), but have been directly measured for CAPABL with a transmitter polarization purity of 123:1 and a receiver polarization purity of $> 800:1$, resulting in  a system bias in the depolarization ratio no greater than 0.8%. This work uses the definition of the backscattering phase matrix as given by Neely et al. (2013) in their Eq. 5. Note that all constant terms of Eq. 1, which will cancel when taking signal ratios, are lumped into the term $\xi(R)$ such as the measurement solid angle, geometric overlap, range resolution, and atmospheric transmission. The transmission Mueller matrix is thus the identity matrix.

Here the number of measured photons incident upon the photodetector, $N_M(R)$, is a function of transmitted and received polarization angle $\phi$ and $\theta$, respectively, and is related to the scattering phase matrix terms, $F_{11}(R)$, $F_{12}(R)$, $F_{22}(R)$, and $F_{33}(R)$ which are all functions of range. For CAPABL, $\phi = 45^o$; applying this constraint to Eq. 2 cancels the functional dependency on $F_{22}(R)$ by design. Thus, using three distinct receiver polarization channels: $\theta_1$, $\theta_2$, and $\theta_3$, one can create a set of three simultaneous equations which can be inverted to calculate the Mueller matrix terms of interest which describe backscattering efficiency, depolarization, and diattenuation (used to determine preferential orientation of scatterers). This set of equations is given in Eq. 3  as

$$\begin{bmatrix} N_1(R) \\ N_2(R) \\ N_3(R) \end{bmatrix} = \xi(R)\begin{bmatrix} 1 & \cos(2\theta_1) & \sin(2\theta_1) \\ 1 & \cos(2\theta_2) & \sin(2\theta_2) \\ 1 & \cos(2\theta_3) & \sin(2\theta_3) \end{bmatrix}\begin{bmatrix} F_{11}(R) \\ F_{12}(R) \\ F_{33}(R) \end{bmatrix} \rightarrow \bar{N} = \bar{\bar{A}}\bar{F}. \tag{3}$$

The general matrix inverse of $\bar{\bar{A}}$ is given in Eq. 4  as

$$\bar{\bar{A}}^{-1} = \frac{1}{\zeta}\begin{bmatrix} \sin(2\theta_2 - 2\theta_3) & \sin(2\theta_3 - 2\theta_1) & \sin(2\theta_1 - 2\theta_2) \\ \sin(2\theta_3) - \sin(2\theta_2) & \sin(2\theta_1) - \sin(2\theta_3) & \sin(2\theta_2) - \sin(2\theta_1) \\ \cos(2\theta_2) - \cos(2\theta_3) & \cos(2\theta_3) - \cos(2\theta_1) & \cos(2\theta_1) - \cos(2\theta_2) \end{bmatrix}. \tag{4}$$

Note that the matrix $\bar{\bar{A}}$ and the matrix inverse $\bar{\bar{A}}^{-1}$ are not functions of range but only of the selected receiver polarizations. The term

$$\zeta = \cos\left(2\theta_3\right)\left(\sin\left(2\theta_2\right) - \sin\left(2\theta_1\right)\right) + \cos\left(2\theta_1\right)\left(\sin\left(2\theta_3\right) - \sin\left(2\theta_2\right)\right) + \cos\left(2\theta_2\right)\left(\sin\left(2\theta_1\right) - \sin\left(2\theta_3\right)\right) \tag{5}$$

5  is introduced in Eq. 4 as a constraint on the validity of the inversion where $\zeta = 0$ results in a degenerate inversion because of receiver polarization selection.

A general form of depolarization

$$d - 1 = \frac{F_{33}}{F_{11}} = \frac{\left(\cos\left(2\theta_3\right) - \cos\left(2\theta_2\right)\right)N_1 + \left(\cos\left(2\theta_1\right) - \cos\left(2\theta_3\right)\right)N_2 + \left(\cos\left(2\theta_2\right) - \cos\left(2\theta_1\right)\right)N_3}{\sin\left(2\theta_2 - 2\theta_3\right)N_1 + \sin\left(2\theta_3 - 2\theta_1\right)N_2 + \sin\left(2\theta_1 - 2\theta_2\right)N_3} \tag{6}$$

and diattenuation

$$D = \frac{F_{12}}{F_{11}} = \frac{\left(\sin\left(2\theta_3\right) - \sin\left(2\theta_2\right)\right)N_1 + \left(\sin\left(2\theta_1\right) - \sin\left(2\theta_3\right)\right)N_2 + \left(\sin\left(2\theta_2\right) - \sin\left(2\theta_1\right)\right)N_3}{\sin\left(2\theta_2 - 2\theta_3\right)N_1 + \sin\left(2\theta_3 - 2\theta_1\right)N_2 + \sin\left(2\theta_1 - 2\theta_2\right)N_3} \tag{7}$$

10  for CAPABL can be expressed in terms of arbitrary observation angles  assuming the condition $\zeta \neq 0$ (for CAPABL $\zeta \approx -2$ calculated from receiver polarizations via atmospheric calibration performed for each measurement). Note that the range dependency of depolarization $(d)$, diattenuation $(D)$, $F_{\#\#}$, and $N_\#$ are dropped to simplify the expressions.

The expressions given in Eq. 6 and Eq. 7 are generalizations of the equations presented by Neely et al. (2013) that assume
15  fixed receiver polarization angles. The diattenuation equations presented by Neely et al. (2013) in their Eq. 7 and Eq. 20 can be recovered from our Eq. 7 by using $\theta_1 = 45^o$, $\theta_2 = -45^o$, and $\theta_3 = 0^o$ for their Eq. 7 and $\theta_1 = 45^o$, $\theta_2 = -45^o$, and $\theta_3 = \pm90^o$ for their Eq. 20. The depolarization term presented by Neely et al. (2013) in their Eq. 8 can be recovered with either set of angles from our Eq. 6. For clarity, retrievals performed with equations from Neely et al. (2013) are referred to as traditional or orthogonal as the polarizations used are orthogonal in Poincare space. The retrievals using Eq. 6 and 7 are referred to as
20  non-orthogonal as they require no such assumption.

Equations 6 and 7 are valid for randomly or preferentially oriented axially symmetric scatterers. If random orientation is observed, diattenuation will be strictly $D = 0$ and the scattering Mueller matrix simplified to a function of two elements, depolarization $d$ and the volume backscatter coefficient $\beta$. This form of the backscatting phase matrix has been given by Mishchenko and Hovenier (1995); Flynn et al. (2007); Gimmestad (2008); Hayman and Thayer (2009, 2012) and is completely
25  consistent with their definitions of depolarization.

**2.2  Dynamic Range**

Polarimetric lidar places stringent requirements on optical detection  system to detect intensity changes between polarization states over a large range of signal strengths, also known as signal dynamic range. The total dynamic range of observed signals arises from many terms
30  in the SVLE. This work will parse two, which will be referred to as range-induced dynamic range and  polarization-induced dynamic range. Range-induced dynamic range arises from the solid angle term, $A/R^2$, and causes signal

strength to vary significantly over the altitude range of interest, especially for tropospheric lidar systems. From the initial signal overlap range of $50\,m$ to $5\,km$, this term changes by four orders of magnitude. This is two orders of magnitude greater than a change from $10\,km$ to $100\,km$.

Polarization-induced dynamic range arises from having different scattered signal intensities from the same scattering volume and detected by different receiver polarization states. Polarization-induced dynamic range is mathematically defined by Eq. 2 by picking different values of $\theta$. An example of this is the frequently used lidar depolarization ratio, which is the ratio of signals measured in cross-polarized and co-polarized channels, referred to throughout this work as perpendicular and parallel, respectively, defined further in Sect. 2.3. A depolarization ratio measurement of 1% indicates parallel and perpendicular polarization signals from the same volume differ by 2 orders of magnitude whereas a measurement of 100% indicates the signals are equal in magnitude.

Combining range-induced and polarization-induced dynamic range to measure a depolarization ratio value of 1% from $50\,m$ to $5\,km$ spans 6 orders of magnitude of signal. This includes observations from the weakest high-altitude perpendicular signal to the strongest low-altitude parallel signal. Figure 1 frames the dynamic range terms for a general set of polarization signals. These data are taken during a relatively clear sky period at Summit. Figure 1 shows raw signals observed from CAPABL using arbitrary units to highlight the dynamic range caused by the measurement of depolarization and by the solid angle term of Eq. 1. In this case, polarization-induced dynamic range introduces approximately 1.5 orders of magnitude of signal within the scattering volume, hereafter referred to as a voxel, and approximately 4.5 orders of magnitude of signal due to range coverage. Adding these together, to measure clear air backscatter intensities from $50\,m$ to $10\,km$ would require an observing system with no fewer than 6 orders of magnitude of signal capacity. Note that clear air and liquid bearing clouds both have high polarization-induced dynamic range, i.e. low depolarization ratio, but liquid bearing clouds have more rapid attenuation of signal, which is not range-induced dynamic range but rather signals reduced by attenuation captured in the transmission terms of the SVLE.

Presented in Fig. 1 are observations from CAPABL using two different observational methods: photon counting and analog detection. Photon counting systems are capable of measuring weak light signals, while analog systems sacrifice sensitivity to measure stronger signals. In photon counting, detector signals are discriminated with a fixed voltage threshold whereas all voltages are summed for analog detection. More detail on the functional differences of photon counting and analog detection is provided in Sect. 3.

As a result of this large dynamic range in signals, the requirement that each receiver modality is acting linearly, i.e. that there is a linear correspondence between incident intensity observed at the receiver photodetector for each polarization state (the information provided by the SVLE) and the number of photons counted, or voltage measured, is critically important. This assumption is known to be limiting for photon counting detection due to the possibility of multiple photons arriving at the detector at the same time (Whiteman et al., 1992; Donovan et al., 1993; Liu et al., 2009; Newsom et al., 2009). Pulse pileup,

referred to throughout this work as saturation, results in the under-representation of signal intensities  by photon counting detection in some polarization states of the observed lidar profile while  other polarization states are unaffected. Critically, saturation can affect different polarization channels in an uneven manner and, therefore, directly cause biases in geophysical retrievals when using signal intensity. In Fig. 1, this is seen clearly as saturation affects the photon
5   counting parallel signal below $6\,km$  and the photon counting perpendicular signal  below $2\,km$.

**2.3   Depolarization Ratio and Saturation**

To fully demonstrate retrieval errors caused by saturation, the traditional  lidar depolarization ratio, $\delta$, defined as:

$$\delta(R) = \frac{S_{0_\perp}(R)}{S_{0_{||}}(R)} = \frac{d(R)}{2 - d(R)} \tag{8}$$

is considered,  in its standard form as well as its relation to depolarization $d$ (Schotland et al., 1971; Intrieri
10 et al., 2002; Flynn et al., 2007; Shupe, 2007; Gimmestad, 2008; de Boer et al., 2009; Hayman and Thayer, 2009, 2011). It should be noted that depolarization $d$ is a Mueller matrix element only (Gimmestad, 2008) resulting from the polar decomposition procedure of Lu and Chipman (1996); Hayman and Thayer (2012); depolarization ratio, $\delta$, has been often related to hydrometeor phase but is not an element of the Mueller formalism directly. Errors in depolarization ratio linked to multiple scattering are well documented (Eloranta, 1998). As Mie scattering only predicts zero depolarization for the backscattering direction
15 (Van De Hulst, 1957; Bohren and Huffman, 1998; Bohren and Clothiaux, 2005), photons that have been multiple scattered by spherical targets can display non-zero depolarization. This work adds analysis of depolarization errors due primarily to saturation.

In Eq. 8, $\delta(R)$ is the lidar depolarization ratio as a function of range, $S_{0_\perp}$ is the photon arrival rate at the detector surface in the perpendicular channel as a function of range, and $S_{0_{||}}$ is the photon arrival rate at the detector surface in the parallel channel as
20 as function of range.  $S_{0_\perp}$ and $S_{0_{||}}$ are the theoretical arrival rates and not the observed arrival rates. If careful attention is not paid to signal saturation, then the stronger parallel signals may be underestimated causing the depolarization  ratio to be overestimated which can be easily mistaken for geophysical signatures. If, for example, a non-paralyzable  saturation model is assumed that links theoretical and observed photon count arrival rates, Eq. 8 can be recast as given
25

as

$$\delta_O(R) = \frac{\frac{S_{0_\perp}(R)}{1 + \tau S_{0_\perp}(R)}}{\frac{S_{0_{||}}(R)}{1 + \tau S_{0_{||}}(R)}} = \delta(R) \frac{1 + \tau S_{0_{||}}(R)}{1 + \tau S_{0_\perp}(R)} \tag{9}$$

to link the theoretical depolarization ratio to the observed depolarization ratio. Note that the non-paralyzable model assumes that it takes some finite time for the photon counting system to reset before it can count another photon and is correct for
30 CAPABL's photon counting system based on the analysis and further description provided in Appendix A.

Here $\delta_O(R)$ is the observed depolarization ratio and $\tau$ is the photon counting system dead time described by Donovan et al. (1993). CAPABL's photon counting acquisition is best modeled as a non-paralyzable system with a dead time of approximately $6\,ns$, therefore, depolarization ratio values observed from parallel channel count rates exceeding 1 to $10\,MHz$ are noticeably biased high. Qualitatively, this effect can be seen in Fig. 1 where the signal induced dynamic range is virtually constant for

5    analog detection but a strong function of height for photon counting detection.

Depolarization ratio error is shown quantitatively in Fig. 2 for many possible ways of measuring depolarization. A similar procedure is performed as in Fig. 1 where photon count rates are modeled from the SVLE and then used in the retrievals given by Eq. 9, but starting with Eq. 6, and applying a combination of receiver polarization angles into the depolarization ratio calculation. This is done for 6 sets of polarization angles, roughly equivalent to those measured by CAPABL, to demonstrate

10    the biases inherent in the possible depolarization measurements. The traditional way of measuring depolarization requires parallel and perpendicular signals which maximizes the  polarization-induced dynamic range, given in panel (a) of Fig. 2. Panels (b) through (f) show possible alternatives that either show less sensitivity to saturation or more uniform sensitivity to saturation using receiver polarizations other than the standard parallel and perpendicular. Using the threshold of  $\delta_O = 0.11$ defined by Intrieri et al. (2002); Shupe (2007), these biases can, at high count rates, exceed the limit set

15    between liquid and ice making it impossible to observe liquid water even if the true depolarization ratio is smaller than the set threshold, i.e. $\delta_O(R) > 0.11$ when $\delta(R) \leq 0.11$.  The effect of saturation shown in panel (a) is neither uniform with count rate or true depolarization ratio nor is it negligibly small relative to the limit set between liquid and ice. The alternatives in panels (b) through (f) offer more uniformity and  reduced bias.

In the polar regions, given that most liquid clouds are relatively low-lying, optically thick, and occur in all seasons, saturation

20    will affect signal levels frequently for photon counting detection (Intrieri et al., 2002; Turner, 2005; Shupe et al., 2006; de Boer et al., 2009; Shupe et al., 2011; Shupe, 2011; Shupe et al., 2013; Cesana et al., 2012).  This saturation will directly bias depolarization values, which will ultimately cause the misrepresentation of liquid clouds as ice clouds

25    . The results of this paper quantify the extent of the saturation impact for CAPABL for a 4 month period from July 2015 to October  2015 and demonstrates the improvements in cloud characterization when using multiple polarization planes.

**2.4   Diattenuation**

As discussed, observed depolarization ratios are a function of atmospheric scattering, optical system setup, and recording

30    systems. Traditional two-channel polarization systems can not unambiguously measure atmospheric depolarization without additional information. However, separating atmospheric depolarization from systematic effects is non-trivial. Hayman and Thayer (2009) show, for example, how to remove depolarization ratio effects caused by receiver optical retardance and scattering. However, recording systems that are subject to saturation can also cause depolarization ratio effects, which are not constant in range and can not be calibrated using methods like that presented in Hayman and Thayer (2009).

The CAPABL system requires at least 3 polarization measurements to measure the $F_{11}(R)$, $F_{12}(R)$, and $F_{33}(R)$. However, saturation has been observed to cause biases in CAPABL measurements using only 3 polarizations. Thus, a forth polarization channel is added, three to measure atmospheric properties and one to monitor recording system effects. Specifically, the $F_{12}(R)$ term is measured twice using two sets of polarization channels with opposite sensitivity to saturation. If the $F_{12}(R)$ terms measured in two different ways are consistent at a given altitude, the lidar counting system is operating normally. An advantage of this over-constrained polarization retrieval is that CAPABL can actively monitor if the polarization measurements are acting properly or are causing systematic biases. A combination of any 3 of the 4 polarization channels can be used to optimize CAPABL's retrievals if the polarization signals are not subject to saturation. If $F_{12}(R)$ is zero, i.e no preferentially oriented ice is present, only 2 of the 4 channels are needed. However, if the polarization retrievals are acting improperly, subject to saturation, CAPABL can identify measurements with non-physical retrieved values and separate them from geophysical values.

[revised manuscript text omitted]

By design, CAPABL uses 4 polarization channels to measure 3 elements of the scattering Mueller matrix: $F_{11}$, $F_{12}$, and $F_{33}$ with one additional measurement to monitor saturation. If saturation is not an issue, any 3 of the 4 channels may be used for the inversion of polarization properties. The utility of the generalization of the polarization theory of Neely et al. (2013) for this work is that the 3 signals with the least error can be used at any time. For example, the 3 strongest signals for measurements of high ice clouds where backscattered signals are weaker or the 3 weakest measurements for low liquid clouds where the backscattered signal is stronger.

**4.2  Automatic Algorithm Bounds**

Using all of this informationthe polarization processing listed above, the masking of data is performed in the following manner. Clear air is found as any time and altitude bin, referred to here as a pixelvoxel, with a backscattering ratio less than 26. Subvisible clouds and aerosols are any pixel voxel with a backscattering ratio between 26 and 50. Clouds are tagged as pixels voxels with backscattering ratio greater than 50. Within cloud pixelsvoxels, the depolarization ratio threshold, originally defined by Intrieri et al. (2002) of $\delta \geq 0.11$ $\delta_Q \geq 0.11$ was used to define ice and $\delta < 0.11$ $\delta_Q < 0.11$ as water. As the most common aerosol at Summit is ice, any pixels voxels tagged as aerosol that displays a depolarization $\delta \geq 0.11$ ratio $\delta_Q \geq 0.11$ is reset as ice. Finally, preferentially oriented ice crystals are identified by $D_1 D_2 > 0.01$ with $\sigma_{D_1}, \sigma_{D_2} \leq 0.05$.

**4.3**

[revised manuscript text omitted]

    There is a dramatic underestimate of liquid water by CAPABL's PC acquisition, which worsens with decreasing altitude, shown in Fig. 5. At 1000 m, PC and A differ by 94% (PC observes 34  voxels of water over the month and analog observes 544). This difference is attributed to liquid  voxels being mistaken for ice due to saturation in the PC

20  parallel channel causing erroneously high depolarization ratio values. Below 2.5 km orthogonal and non-orthogonal results are nearly identical but above that altitude non-orthogonal polarization retrievals see more clear air and ice and have higher rates of effective sampling due to the stronger signals used. Recall the 3rd channel measurement is used in the non-orthogonal calculation, which is stronger than  perpendicular measurements and reduces the  polarization-induced dynamic range between the 3rd channel measurement and the  parallel measurement allowing for greater

25   range coverage.

[revised manuscript text omitted]

**6.2**

25

30

**6.2 Radiative Implications of Cloud Phase Misidentification**

Cloud phase is shown to be an important driver of the radiation budget in the Arctic (Curry et al., 1996; Miller et al., 2015, 2017). For a constant amount of water, the liquid phase affects the radiative budget more strongly because it tends to form many
25  small droplets with larger surface area where ice tends to form fewer larger crystals. As such, the optical depth and longwave emission of liquid emissive flux by liquid clouds is greater than of ice. At Summit, Miller et al. (2015) show that cloud radiative forcing is driven primarily by the annual variability in liquid bearing clouds. It is therefore important to assess and optimize the CAPABL data in terms of the fractional occurrence of liquid, ice, and clear air states.

Figure 7 gives the fractional occurrence of pixel voxel types in the column above CAPABL. Profiles are defined from the
30  ground to 15 km for a given time by the radiatively dominant pixel voxel type, providing a vertically integrated snapshot of the profile above Summit. For example, if a profile contains liquid water, ice, aerosol, and and ice or clear air, it is defined as liquid. Ice is defined as any column with ice pixels in it but voxels in it that lacks liquid. Sub-visible columns must contain that pixel type without ice or liquid cloud pixels present. Clear air columns must contain nothing but clear air. In this way, one can convert the pixel voxel number defined in Fig. 6 to a cloud fraction proxy fractional occurrence.

An examination of Fig. 7 suggests that all 6 processing types: A, PC, SCPC for both orthogonal and non-orthogonal, yield  fractional occurrence differences of clear air  voxels (differences of 13%, 9%, 6%, and11% for July, August, September, and October, respectively). This is due largely to the observational sensitivity of the separate channels, analog orthogonal is the least sensitive, i.e. misses the occurrence of high altitude clouds, and non-orthogonal photon counting is the most sensitive. Figure 7 also shows  an increase in that difference by a factor of approximately 3 in liquid column identification (differences of 41%, 31%, 28% and 15% for July, August, September, and October, respectively, for A and PC). This difference in liquid water cloud fraction can be used to approximate an error in cloud radiative forcing using the results from Fig. 7 from Miller et al. (2015). Using an  difference of 30%, this time period of fractional occurrence of liquid clouds equates to an error in longwave cloud radiative forcing of approximately $10\,W/m^2$. Miller et al. (2015) finds an average of $33\,W/m^2$ for cloud radiative forcing at Summit suggesting that using uncorrected CAPABL data to infer radiative impacts could under-represent forcing by as much as one third.

**6.3 Identifying Saturation**

One major finding from analysis of 5 years of CAPABL data is the sensitivity of the diattenuation measurement to saturation. One can see clearly in Fig. 4 that the areas identified as having erroneous depolarization measurements have diattenuation values, $D_1 \cdot D_2$ in excess of -0.05. The combined diattenuation product, described in Sect. 4.1 and Table 2 in step 10, can distinguish between preferential orientation and saturation. For the 4 month data period examined, voxels identified as ice by photon counting and liquid by analog detection have an average diattenuation combination, $D_1 \cdot D_2$, of -0.08 or an average value $|D_1| = |D_2| = 0.28$. This is a result of an over constraint on the retrieval of 3 polarization properties, $F_{11}$, $F_{12}$, and $F_{33}$, with 4 polarization channels. With this diattenuation combination, CAPABL is able to self analyze data that is erroneous due to the counting system's inability to linearly measure all polarizations. This identification is not possible with a polarimetric lidar with 2 channels without additional information.

**6.4 Recognized Future Work**

For certain designs, consideration of the signal dynamic range may be as important as the selection of polarization planes. The same problems related to dynamic range that are demonstrated for CAPABL likely exist in other one detector designs, like the polarization sensitive MPL, because the  polarization signals can still vary by as much as two orders of magnitude  due to polarization-induced dynamic range for very low depolarization ratio targets like liquid-only clouds, mixed-phase clouds, and clear air. More work is suggested to determine the possible errors in single detector designs as they are often designed to use orthogonal polarizations, which have  large polarization-induced dynamic ranges for liquid clouds and clear air.

[revised manuscript text omitted]

Lu, S.-Y. and Chipman, R. A.: Interpretation of Mueller matrices based on polar decomposition, J. Opt. Soc. Am. A, 13, 1106–1113, doi:10.1364/JOSAA.13.001106, http://josaa.osa.org/abstract.cfm?URI=josaa-13-5-1106, 1996.

Measures, R. M.: Laser Remote Sensing: Fundamentals and Applications, John Wiley and Sons, 6000 Broken Sound Parkway NW, Suite 300, Boca Raton, FL 33487-2742, USA, 1984.

Miller, N. B., Shupe, M. D., Cox, C. J., Walden, V. P., Turner, D. D., and Steffen, K.: Cloud Radiative Forcing at Summit Greenland, Journal of Climate, 28, 6267–6280, 2015.

Miller, N. B., Shupe, M. D., Cox, C. J., Noone, D., Persson, P. O. G., and Steffen, K.: Surface Energy Budget Responses to Radiative Forcing at Summit, Greenland, The Cryosphere, 11, 497–516, doi:10.5194/tc-11-497-2017, 2017.

Mishchenko, M. I. and Hovenier, J. W.: Depolarization of light backscattered by randomly oriented nonspherical particles, Opt. Lett., 20, 1356–1358, doi:10.1364/OL.20.001356, 1995.

[revised manuscript text omitted]
  ice  and water voxels.  as ice, a column must contain ice but lack water voxels. If a column contains a water voxel, the column is labeled as liquid. The fractional occurrences are given for each bar rounded to the nearest thousandth.

[Figure]

**Figure 8.** Saturation analysis of the CAPABL photon counting channel using the theory developed by Donovan et al. (1993); Whiteman (2003); Liu et al. (2009). The ideal signal count rate is found by normalizing the analog detection channel to the photon counting channel in a region where both are acting linearly which is about 500 kHz count rate. The measured count rate is then taken directly from photon counting measurements. The paralyzable and non-paralyzable models are then fit using a Levenberg-Marquardt weighted non-linear least squares fitting algorithm of the observed calibration data. The $1\sigma$ confidence bound is given for each dead time fit parameter. Finally, the percent error of the correction model is given relative to the ideal count rate on the right ordinate as diamonds.

**Table 1.** CAPABL current system specifications. Note that polarization purity and polarization rejection are measured quantities. Polarization purity is measured with a 100,000:1 Glan-Taylor polarizer.

| Transmitter | Receiver | Signal Processing |
|---|---|---|
| Big Sky Laser Ultra flashlamp pumped Nd:YAG | Schmidt Cassegrain Telescope | Combined Analog and Photon Counting acquisition |
| Wavelength: 532.3 nm | Receiver Aperture: 20.8 cm | Data system: |
| Pulse Energy: 60 mJ | Filter Bandwidth: 0.3 nm | Licel Transient Recorder TR20-12 Bit |
| Pulse Rate: 15 Hz | Channels: 1 | Range bin size: 7.5 m |
| Twin Head | Field of View: 1.4 mrad | Integration time: 5 sec |
| Polarization Purity: $> 123 : 1$ | Polarization Rejection: $> 800 : 1$ | PMT: Hamamatsu R7400U-03 |
| | Linear Polarizations Observed: 4 | |

**Table 2.** A summary of the data processing steps taken to create the data masks desired for CAPABL. The processing for each data type: Analog (An), Photon Counting (PC), and Saturation Corrected Photon Counting (SCPC), is constant except where noted. Note that the diattenuation error equation is calculated per standard propagation of error techniques taking a Taylor series expansion of Equation 7.

| | Processing Step | Channels | Details |
|---|---|---|---|
| 1) | Time integration | An/PC | To a constant 20 second resolution |
| 2) |  Spatial integration | An/PC | To a constant 30 meter resolution |
| 3) | Saturation correction | PC | Creates SCPC level |
| 4) | SNR filter | All | |
| 5) | Speckle filter | All | $5 \times 5$ surrounding box |
| | | | $> 75\%$ data already removed $=$ bad |
| | | | $> 25\%$ data available $=$ good |
| 6) | Calculate polarization properties | All | Depolarization and depolarization ratio per Eq. 6 and 8 |
| | | | Depolarization and depolarization ratio error per error propagation of Eq. 6 and 8 |
| | | | Diattenuation per Eq. 7 |
| | | | Diattenuation error per error propagation of Eq. 7 |
| | | | Backscatter ratio ($R$) per (Klett, 1981; Neely et al., 2013) |
| 7) | Remove non-physical values | All | Values outside  $0 \leq \delta_Q \leq 1$ |
| | | | Values outside  $0 \leq \sigma_{\delta_Q} \leq 0.4$ |
| | | | Values outside $-1 \leq D \leq 1$ |
| | | | Values outside $0 \leq \sigma_D \leq 0.2$ |
| 8) | Calculate base mask | All | Clear: $1 \leq R < 26$ |
| | | | Aerosol: $26 \leq R < 50$ |
| | | | Cloud: $R \geq 50$ |
| 9) | Calculate phase mask | All | Liquid: cloud  voxels with $0 \leq \delta_Q \leq 0.11$ |
| | | | Ice: cloud  voxels with $\delta_Q > 0.11$ |
| 10) | Calculate orientation mask | All | Random: ice with $0 \leq D_1 D_2 \leq 0.01$ |
| | | | Preferential: ice with $D_1 D_2 \geq 0.01$ and $\sigma_D \leq 0.05$ |